# HASTY, the *Arabidopsis EXPORTIN5* ortholog, regulates cell-to-cell and vascular microRNA movement

Florian Brioudes ⓘD, Florence Jay, Alexis Sarazin, Thomas Grentzinger, Emanuel A Devers & Olivier Voinnet\* ⓘD

## Abstract

Plant microRNAs (miRNAs) guide cytosolic post-transcriptional gene silencing of sequence-complementary transcripts within the producing cells, as well as in distant cells and tissues. Here, we used an artificial miRNA-based system (amiRSUL) in *Arabidopsis thaliana* to explore the still elusive mechanisms of inter-cellular miRNA movement *via* forward genetics. This screen identified many mutant alleles of *HASTY* (*HST*), the ortholog of mammalian *EXPORTIN5* (*XPO5*) with a recently reported role in miRNA biogenesis in *Arabidopsis*. In both epidermis-peeling and grafting assays, amiRSUL levels were reduced much more substantially in miRNA-recipient tissues than in silencing-emitting tissues. We ascribe this effect to HST controlling cell-to-cell and phloem-mediated movement of the processed amiRSUL, in addition to regulating its biogenesis. While HST is not required for the movement of free GFP or siRNAs, its cell-autonomous expression in amiRSUL-emitting tissues suffices to restore amiRSUL movement independently of its nucleo-cytosolic shuttling activity. By contrast, HST is dispensable for the movement and activity of amiRSUL within recipient tissues. Finally, HST enables movement of endogenous miRNAs that display mostly unaltered steady-state levels in *hst* mutant tissues. We discuss a role for HST as a hitherto unrecognized regulator of miRNA movement in relation to its recently assigned nuclear function at the nexus of *MIRNA* transcription and miRNA processing.

**Keywords** *Arabidopsis*; ARGONAUTE; HASTY; miRNA; movement
**Subject Categories** Membranes & Trafficking; Plant Biology; RNA Biology
The EMBO Journal (2021) 40: e107455

## Introduction

In RNA silencing, small (s)RNAs, 20–32-nt in size, incorporate into ARGONAUTE(AGO)-like proteins to guide RNA-induced silencing complexes (RISCs) to sequence-complementary target RNAs (Ghildiyal & Zamore, 2009). In many organisms, silencing sRNAs are preponderantly processed from longer double-stranded (ds)RNA by RNaseIII proteins in the Dicer family. In plants, all known silencing sRNAs are produced by such Dicer-like (DCL) enzymes, of which there are four paralogs (DCL1- > 4) in the model *Arabidopsis* (Bologna & Voinnet, 2014). Among these, DCL3 produces the largest bulk of sRNAs with a signature size of 24 nucleotides. These so called heterochromatic small interfering RNAs (siRNAs) incorporate AGO4, AGO6, and AGO9—three of 9 functional AGOs in *Arabidopsis*—to initiate *de novo* DNA methylation and chromatin compaction at loci encoding transposons and repeats, often resulting in transcriptional gene silencing (TGS) (Matzke *et al*, 2015). Post-transcriptional gene silencing (PTGS), on the other hand, involves the remaining bulk of *Arabidopsis* sRNAs and is exemplified by antiviral defense mediated by virus-derived 21-nt and 22-nt siRNAs, respectively, produced by DCL4 and DCL2 (Lecellier & Voinnet, 2004; Bouche *et al*, 2006; Deleris *et al*, 2006; Ding & Voinnet, 2007). DCL1-dependent micro (mi)RNAs, which represent the second largest bulk of silencing sRNAs in plants, engage into endogenous, as opposed to defensive, PTGS thereby enabling the regulation of many biological processes including developmental patterning and adaptation to stress (Voinnet, 2009). Unlike siRNAs produced as large populations by DCL2, DCL3, and DCL4, plant miRNAs accumulate as discrete species mostly with a 5'-uridine (5'-U). This feature predisposes them to load mainly into AGO1, their cognate effector protein (Mi *et al*, 2008), as part of microRNA-RISCs (miRISCs). In the cytosol, miRISCs execute silencing of endogenous transcripts displaying complementary miRNA-target sites in their coding sequence or UTRs. Silencing occurs via endonucleolytic cleavage, or "slicing" (Baumberger & Baulcombe, 2005) and/or translational repression possibly coupled to accelerated mRNA decay (Brodersen *et al*, 2008; Guo *et al*, 2010; Bazzini *et al*, 2012; Li *et al*, 2013). Under certain circumstances, miRNA-target interactions may be accompanied by the cytosolic *de novo* conversion of target transcripts into dsRNA, which is then processed by DCL4 into populations of 21-nt siRNAs known as trans-acting siRNAs (tasiRNAs) (Vazquez *et al*, 2004b; Gasciolli *et al*, 2005) and phased siRNAs (phasiRNAs) (Fei *et al*, 2013). Both types of siRNAs endow

Department of Biology, ETH Zürich, Zürich, Switzerland
\*Corresponding author. Tel: +41 44 633 93 60; E-mail: olivier.voinnet@biol.ethz.ch

amplified—and sometimes non-cell-autonomous—silencing of *trans* targets in *Arabidopsis*, in an AGO1-dependent manner (Chitwood *et al*, 2009; Schwab *et al*, 2009).

While the silencing activity of miRNAs occurs in the cytosol, their biogenesis and maturation are nuclear processes (Fang & Spector, 2007; Achkar *et al*, 2016). Most plant *MIRNA* genes are independent transcription units yielding non-coding primary transcripts (pri-miRNAs) invariably containing a stem-loop section that is subjected to stepwise processing (Voinnet, 2009). Nuclear DCL1 and cofactors including the dsRNA binding protein HYPONASTIC LEAVES1 (HYL1) and SERRATE (SE) excise the stem-loop from the pri-miRNA to generate a pre-miRNA from which a mature miRNA is then produced via one or several subsequent cuts mediated by DCL1 (Achkar *et al*, 2016). The 3' ends of miRNAs then undergo 2'-O-methylation mediated by HUA ENHANCER1 (HEN1) in the nucleus, which protects them from poly-U tailing and subsequent trimming/degradation by exonucleases (Li *et al*, 2005; Achkar *et al*, 2016). For years, the steps downstream of 2'-O-methylation in the plant miRNA pathway have been mostly inferred from parallels drawn with the animal pathway. In animals, pre-miRNAs—as opposed to processed miRNAs in plants—are exported to the cytosol by EXPORTIN5 (XPO5) for final processing and loading into cognate AGO effectors (Lund *et al*, 2004). The XPO5 *Arabidopsis* ortholog is named HASTY (HST) owing to the accelerated juvenile-to-adult phase transition observed in the namesake mutant, alongside aberrant leaf development, defective phyllotaxis, and sterility (Telfer & Poethig, 1998; Bollman *et al*, 2003). In rice, the HST ortholog CRD1 is additionally required for proper crown root development (Zhu *et al*, 2019). Attempts to implicate HST in nuclear export of plant miRNAs have been unsuccessful, however, since no overt changes are observed in miRNA nucleo-cytosolic partitioning between wild-type (WT) and *hst-1* loss-of-function *Arabidopsis* (Park *et al*, 2005; Bologna *et al*, 2018; Zhang *et al*, 2020; Cambiagno *et al*, 2021). Instead, the steady-state accumulation of ~ 30% miRNAs is reduced, albeit at substantially varying extents, in the *hst* background whereas that of the others is unchanged or occasionally even higher in certain tissues for reasons that have so far eluded investigation (Park *et al*, 2005; Cambiagno *et al*, 2021). Similar observations were made in *crd1* in rice, in which a larger fraction of miRNAs displays reduced levels, albeit, yet again, to greatly varying extents (Zhu *et al*, 2019).

The long-standing idea that, like their counterparts in animals, processed plant miRNAs are mainly loaded into AGO1 in the cytosol was recently challenged by the finding that AGO1 is a nucleocytoplasmic shuttling protein (Bologna *et al*, 2018). This and other observations prompted a revised model of the plant miRNA pathway in which *de novo* translated apo- (i.e., non-loaded) AGO1 undergoes NLS-dependent nuclear import enabling its loading with neo-synthesized, processed miRNAs. The nuclear loading likely exposes an AGO1 nuclear export signal (NES) normally buried inside the non-loaded protein, granting, in turn, translocation of AGO1:miRNA complexes to the cytosol in a manner antagonized by Leptomycin B (Bologna *et al*, 2018), an inhibitor of EXPORTIN1 (XPO1/CRM1). The same work showed that, by contrast, tasi- and phasiRNA loading likely involves the cytosolic pool of AGO1, before its NLS-dependent nuclear import, consistent with tasi-/phasiRNA biogenesis, unlike miRNA biogenesis, being an entirely cytoplasmic process (Bologna *et al*, 2018). More recent and subsequent work

added further ground to the proposed revised model of the plant miRNA pathway by identifying the TREX-2 complex as a likely component of the machinery that exports neo-formed AGO1-miRISCs, from the nucleus to the cytosol (Zhang *et al*, 2020). Indeed, and unlike in *hst* mutants, the cytoplasmic/nuclear ratios of miRNAs are reduced in mutant *Arabidopsis* defective for a TREX-2 core subunit that copurifies with XPO1A/B in mass spectrometry analyses (Zhang *et al*, 2020). Noteworthy, neither the Bologna *et al* (2018) nor the Zhang *et al* (2020) study identified any specific role for HST in either miRNA- or AGO1-nuclear export, a role in fact never claimed in the original study of HST conducted by Park *et al* (2005). Consequently, both the function(s) of HST and the aforementioned molecular effects of the *hst* mutation had remained completely mysterious. This was until a very recent in-depth analysis revealed that *Arabidopsis* HST modulates miRNA biogenesis by interacting with and scaffolding DCL1 recruitment to genomic *MIRNA* loci during pri-miRNA transcription/processing (Cambiagno *et al*, 2021). The same study also confirmed that nucleo-cytosolic shuttling of HST plays no apparent role in nuclear miRNA export.

A fascinating yet poorly understood feature of plant miRNAs is that some of these molecules can act non-cell autonomously (Liu & Chen, 2018). Recent work conducted in the *Arabidopsis* root tip revealed that this property might be more frequent than was originally anticipated (Voinnet, 2009; Brosnan *et al*, 2019), even though the number of documented examples of cell-to-cell movement remains scarce owing, mostly, to the technical challenges posed by such studies. Through the use of homo- or hetero-grafts, more evidence exists supporting long-distance movement of silencing mediated by endogenous as well as artificial miRNAs, although no single miRNA has been studied under both cell-to-cell and long-distance premises thus far (Pant *et al*, 2008; Chitwood *et al*, 2009; Kawashima *et al*, 2009; Buhtz *et al*, 2010; Carlsbecker *et al*, 2010; de Felippes *et al*, 2011; Skopelitis *et al*, 2018; Tsikou *et al*, 2018; Brosnan *et al*, 2019; Han *et al*, 2020; Li *et al*, 2021). While the results of some of the above studies converge in suggesting that processed miRNAs, as opposed to pri/pre-miRNA precursors, are generally involved, the precise molecular form(s) of the mobile entities remains unclear (Pant *et al*, 2008; Ham & Lucas, 2017; Liu & Chen, 2018; Brosnan *et al*, 2019), as do the channels employed for movement. As proposed for siRNA populations (Voinnet *et al*, 1998; Kobayashi & Zambryski, 2007; Rosas-Diaz *et al*, 2018; Devers *et al*, 2020), symplasmic movement has been advocated in certain cases of miRNA movement (Skopelitis *et al*, 2018; Brosnan *et al*, 2019), which should involve the plasmodesmata (PDs) connecting many plant cells including the phloem sieve elements (SEs) (Yan & Liu, 2020). Accordingly, PD obstruction via inducible callose deposition was shown to impede miR165/166 endodermis-> stele mobility in the *Arabidopsis* root tip in a manner requiring the PD-associated BAM1/2 receptor-like kinases (RLKs) (Vaten *et al*, 2011; Fan *et al*, 2021). Whether this single example can be extrapolated to other miRNAs remains unknown, however, especially given that additional studies conducted in the same organ uncovered intriguing miRNA movement patterns that are difficult to reconcile with mere symplasmic spread (Brosnan *et al*, 2019). Alternatively, or additively to symplastic transport, an active apoplastic pathway also mediates photo-assimilates' phloem loading in some plant species including *Arabidopsis* (Gahrtz *et al*, 1994). Furthermore, possibly secreted exosome-like vesicles underpin host-induced gene silencing

mediated by *Arabidopsis* mi- and siRNAs during interactions with a necrotrophic fungus, presumably via the apoplast (Cai *et al*, 2018). Whether exosomes or other vesicles are used in sRNA trafficking between plant cells remains unknown, however, especially because plant cells are separated by cell walls unlikely to accommodate the passage of vesicles. As proposed recently, miRNA movement may well involve multiple parallel channels or combinations thereof (Liu & Chen, 2018). Last but not least, intracellular mechanisms regulating miRNA movement, if any, remain to be discovered. Such mechanisms could potentially help explaining why only some, unlike other miRNAs, display non-cell-autonomous activities or why any given miRNA might act non-cell autonomously only in certain tissues unlike others (Parizotto *et al*, 2004; Voinnet, 2009; Schulze *et al*, 2010; Brosnan *et al*, 2019; Han *et al*, 2020).

Here, we report how a forward genetic screen for *Arabidopsis* mutants impaired in movement of an artificial miRNA (amiRSUL) identified a vast collection of *hst* alleles. Our results suggest that HST is cell autonomously required for facilitating both cell-to-cell movement and phloem-based long-distance movement of the fully processed amiRSUL, but also of all endo-miRNAs tested in our study. The requirement for HST appears independent of its nucleo-cytoplasmic shuttling and is asymmetrical between tissues, with the protein being necessary for amiRSUL emission in silencing-emitting tissues, but dispensable for its reception in silencing-recipient tissues. These and additional findings are discussed in the context of a recently discovered scaffolding role for HST linking *MIRNA* transcription and biogenesis in the nucleus (Cambiagno *et al*, 2021).

## Results

### amiRSUL-mediated silencing moves between cells and across organs via the phloem

*Arabidopsis* pri-miR319 was re-engineered into 5'U-terminal amiRSUL predicted to target, in an AGO1-dependent manner, the magnesium chelatase subunit *CHLORINA42* (*CH42* or *SUL*) mRNA, which is required for chlorophyll accumulation (Fig 1A; Appendix Fig S1A) (de Felippes *et al*, 2011). *pri-amiRSUL* was mobilized under the phloem companion-cell (CC)-specific *pSUC2* promoter to produce several *pSUC2::amiRSUL* single-locus transgenic lines. All displayed vein-proximal chlorosis (Fig 1B), the expected phenotypic output of *SUL* silencing (de Felippes *et al*, 2011). Accordingly, the *SUL* mRNA levels were lower in *pSUC2::amiRSUL* compared to non-transgenic leaves (Appendix Fig S1B). Chlorosis was impaired by mutations compromising miRNA, but not siRNA biogenesis (Appendix Fig S1C). It was also reduced by the hypomorphic *ago1-27* mutation (Morel *et al*, 2002), which produces a partially functional protein due to a PIWI domain lesion (Mallory *et al*, 2009), thus confirming that, like endo-miRNA-mediated silencing, amiRSUL silencing operates via AGO1. In parallel, we produced *pSUC2::GUS* transgenic plants carrying a transgene engineered with the same *pSUC2* promoter fragment used in *pSUC2::amiRSUL* plants. Despite the high sensitivity of the method (Jefferson *et al*, 1987), histochemical GUS staining of whole *pSUC2::GUS* seedlings confirmed no detectable ectopic or spurious activity of *pSUC2* aside from its previously reported, highly CC-specific expression pattern (Fig 1C) (Truernit & Sauer, 1995). The fact that

chlorosis in *pSUC2::amiRSUL* leaves manifested beyond the CC-restricted activity of *pSUC2* thus suggested amiRSUL-silencing movement. This notion was directly confirmed by crossing *pSUC2::amiRSUL* with a previously reported *Arabidopsis* line expressing a membrane-anchored, i.e., non-mobile, allele of GFP (tmGFP9) under the *pSUC2* promoter (Stadler *et al*, 2005), referred to as *pSUC2::tmGFP9* thereafter. Overlaying the chlorotic and GFP patterns indeed revealed that the former extends several cells beyond the latter (Appendix Fig S2), indicating that amiRSUL silencing moves from the CCs to neighboring cells in the leaves' lamina.

The *SUL* mRNA levels were reduced in non-transgenic WT rootstocks micro-grafted onto *pSUC2::amiRSUL* scions compared with those in non-transgenic WT rootstocks micro-grafted onto non-transgenic WT scions (Fig 1D), suggesting that amiRSUL silencing also moves between organs over long distances. Accordingly, amiRSUL was readily detected in non-transgenic WT rootstocks micro-grafted onto *pSUC2::amiRSUL* WT scions (Fig 1E), and, strikingly, its accumulation was comparable with that seen in *pSUC2::amiRSUL* WT rootstocks grafted onto non-transgenic WT scions, unraveling a highly efficient long-distance transport. amiRSUL levels were the highest, and those of *SUL* mRNA lowest, in *pSUC2::amiRSUL* rootstocks grafted onto *pSUC2::amiRSUL* scions as expected from the cumulation of the mobile and rootstock-intrinsic pools of amiRSUL (Fig 1D and E). Beside its debated physiological relevance (Buhtz *et al*, 2010), *Arabidopsis* micrografting can generate non-vascular connections enabling reiterated cell-to-cell movement potentially misconstrued as long-distance phloem translocation (Liang *et al*, 2012). To address whether amiRSUL silencing moves over long distances within the phloem sieve elements (SEs), we used aphids, which selectively and non-invasively feed into SEs as opposed to any other cells present in vascular bundles, CCs included (Dixon, 1998). amiRSUL indeed accumulated in aphids fed onto *pSUC2::amiRSUL-*, but not onto non-transgenic plants (Fig 1F). In line with previous indirect observations made with sampling of pumpkin and *Brassica napus* phloem exudates (Yoo *et al*, 2004; Buhtz *et al*, 2008; Buhtz *et al*, 2010), these results directly demonstrate that long-distance movement of amiRSUL silencing proceeds via the phloem stream.

### Movement involves the processed amiRSUL, not its precursors

In principle, movement could involve pri-, pre- and/or fully processed amiRSUL. To distinguish between these possibilities, we grafted *pSUC2::amiRSUL* scions onto non-transgenic rootstocks with either a WT or mutant background. Functional graft transmission of pri- or pre-amiRSUL would require processing and subsequent stabilization, in recipient cells, of mature amiRSUL via 2'-O-methylation mediated by HEN1 (Li *et al*, 2005). We used a fertile, mutant allele of *HEN1*, *hen1-14*, which was isolated by forward genetics as part of the present study (see next section). As expected, amiRSUL levels were substantially reduced and displayed signs of 2'O-uridylation-mediated tailing in *pSUC2::amiRSUL*[hen1-14] plants (Fig 1G). Yet, upon grafting, accumulation of the processed amiRSUL was unchanged in non-transgenic *hen1-14* compared with non-transgenic WT rootstocks (Fig 1G), in line with previous conclusions drawn from graft-transmitted endo-miR399 and endo-miR395 (Buhtz *et al*, 2010). Similar findings were made with non-transgenic rootstocks lacking HYL1 activity required, among other functions,

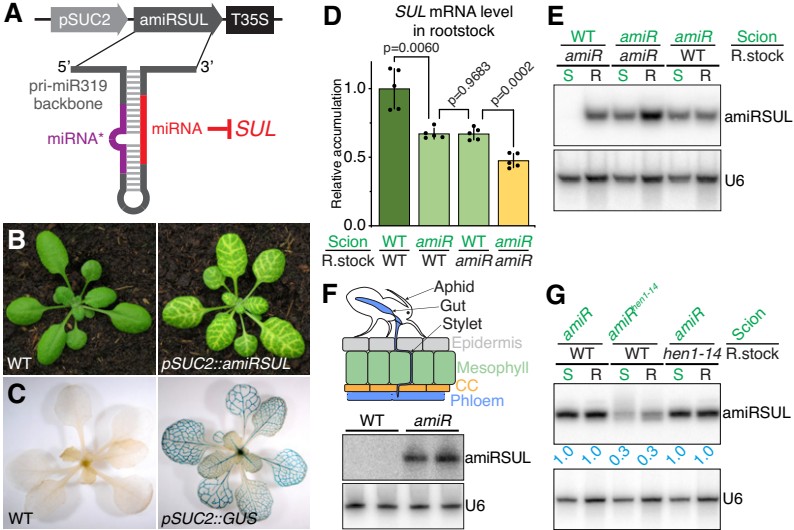

**Figure 1. amiRSUL moves intercellularly and over long distances via sieve elements.**

A  Schematized *pSUC2::amiRSUL* reporter. T35S: CaMV 35S terminator.

B  amiRSUL-silencing phenotype in *pSUC2::amiRSUL* transgenic versus non-transgenic *Arabidopsis*.

C  GUS-staining pattern in *pSUC2::GUS* transcriptional reporter plants.

D  RT–qPCR analysis of *SUL* mRNA levels in WT or *pSUC2::amiRSUL* (*amiR*) rootstocks grafted onto WT or *amiR* scions. Error bars: SD. Welch's *t*-test P-value is indicated. *n* = 5.

E  amiRSUL northern analysis in WT or *amiR* scions (S) or rootstocks (R) in the indicated grafting combinations. U6 RNA hybridization provides an internal RNA loading control.

F  Top: aphids' feeding stylets selectively reach the phloem sieve elements. Bottom: amiRSUL northern analysis, in biological duplicates, from total RNA extracted 14 days post-infestation from aphids fed on WT or *amiR* plants. U6: as in (E).

G  Same as (E) with *amiR*^hen1-14^ scions and non-transgenic WT or *hen1-14* rootstocks. Average relative U6-normalized band-intensity quantifications from two biological replicates are indicated.

Source data are available online for this figure.

for pri-to-pre-miRNA processing (Appendix Fig S3A); accordingly, pri-amiRSUL was consistently below detection limit in WT non-transgenic rootstocks (Appendix Fig S3B). Because long-distance movement occurs, at least partly, via the phloem, a case could be formally made that mobile pri- or pre-amiRSUL might be processed into mature amiRSUL within the SEs of *pSUC2::amiRSUL* scions, before being unloaded into *pSUC2::amiRSUL*^hen1-14^ rootstocks. We consider this possibility highly unlikely, however, because functional differentiation of SEs entails, early in their development (protophloem stage), the self-destruction of the nucleus among other large organelles that would otherwise obstruct phloem movement. This destruction is accompanied by the release and full degradation, in the cytoplasm, of all nuclear DNA, RNA, and protein content (Zhang *et al*, 2009; Furuta *et al*, 2014), which would include the miRNA processing machinery (Fang & Spector, 2007). Collectively, the above results therefore advocate movement of the fully processed amiRSUL, not of its precursors.

## A forward genetic screen for suppressors of amiRSUL silencing identifies many independent *hst* mutant alleles

A forward genetic screen for amiRSUL-silencing-deficient mutants was conducted in 1,750 offsprings of EMS-mutagenized *pSUC2::amiRSUL* seeds. Alongside a small number of presumptive miRNA biogenesis/activity mutants not further discussed here, the fertile

*hen1-14* missense allele used in the grafting experiments of Fig 1G was isolated based on its altered rosette leaf phenotype resembling that of the strong *hen1-6* (SALK_090960; ecotype Col-0) reference allele (Appendix Fig S4A–C) (Li *et al*, 2005). Among the remaining isolated mutants, two independent individuals with compromised amiRSUL silencing showed phenotype segregations consistent with single, recessive, and nuclear mutations. Whole-genome resequencing and short read mapping identified distinct single-base transitions typically induced by EMS within the genomic sequence of *HST*; accordingly, the mutants were named *hst-25* and *hst-26* (Fig 2A). A suboptimal donor-splice site in *hst-25* (AG- > AA mutation) causes partial retention and alternative splicing of intron 12 (Appendix Fig S5), while *hst-26* carries a Ser- > Phe missense mutation in exon 12. In western analysis, *hst-25* and *hst-26* accumulate, respectively, barely detectable and low HST protein levels (Fig 2B), although *hst-25* accumulates residual mRNA isoforms encoding WT and truncated protein versions of HST (Appendix Fig S5).

The *hst-25* lesion is adjacent to the donor-splice site mutation (gt- > ga) originally identified in the reference *hst-1* allele (Telfer & Poethig, 1998; Bollman *et al*, 2003) in which, we verified, retention and similar remnant alternative splicing of intron 12 causes the HST protein to be below detection levels in western analysis (Appendix Fig S5; Fig 2B), correlating with the previously reported aberrant leaf development, phyllotaxy, and sterility of *hst-1* (Telfer & Poethig, 1998; Bollman *et al*, 2003). These anomalies were still

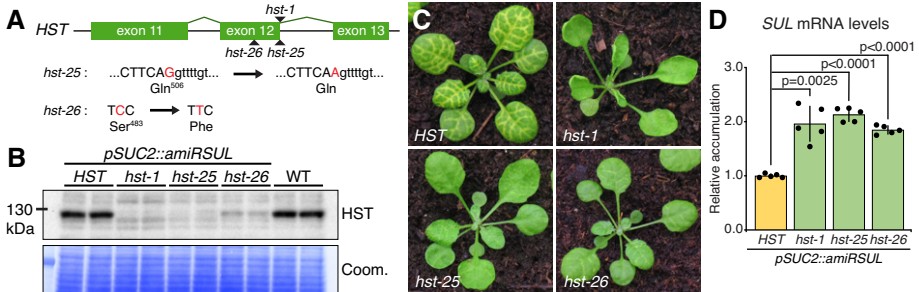

**Figure 2.** *hst* suppresses amiRSUL silencing.

A *HST* genomic sequence spanning exons 11 to 13 and position of the *hst-1/-25/-26* mutations. EMS-induced nucleotide transitions (arrows) in *hst-25* and *hst-26* are in red; upper-/lower-case fonts: exonic/intronic sequences.

B HST western analysis, in biological duplicates, in the indicated genotypes. Coomassie blue staining (Coom.) provides a control for total protein loading.

C Phenotype of *pSUC2::amiRSUL* plants in *hst-1/-25/-26* backgrounds.

D RT–qPCR analysis of *SUL* mRNA levels in leaves with the indicated genotypes. Error bars: SD. Welch's *t*-test *P*-values are indicated. *n* = 5.

Source data are available online for this figure.

apparent, albeit attenuated, in *hst-25* and *hst-26*, both of which displayed strongly reduced, yet not eliminated, vein-centered chlorosis, consistent with hypomorphism (Fig 2C). By contrast, and consistent with stronger developmental defects of the *hst-1* mutant, chlorosis was abrogated by *hst-1* upon its introgression into *pSUC2::amiRSUL* plants (*pSUC2::amiRSUL^hst-1^*; Fig 2C). amiRSUL-silencing deficiency was maintained in F1 progenies of *hst-25/-26* crossed to *pSUC2::amiRSUL^hst-1^* plants (Appendix Fig S6A). amiRSUL silencing was, by contrast, restored in *pSUC2::amiRSUL^hst-1/-25/-26^* plants constitutively expressing a GFP-tagged allele of HST (*pUB10::HST: GFP*; Appendix Fig S6B). These observations genetically confirmed the causality of *hst* for the strongly reduced amiRSUL-silencing phenotype. This prompted us to re-inspect the pool of initially isolated mutants for potential *hst*-like developmental phenotypes. We retrieved at least 13 such additional and independent mutants, all of which displayed vastly reduced HST protein levels; all but three were sufficiently fertile to allow confirmation of their presumptive *hst* mutant genotype via allelism test to *hst-1* (Appendix Fig S7). In total, *hst* alleles accounted for ~ 20% of mutants selected for their reduced vein-chlorotic phenotype in the *pSUC2::amiRSUL* screen, uncovering a striking bias.

### Reduced amiRSUL biogenesis is unlikely to account, alone, for amiRSUL-silencing suppression in *hst*

As expected from their phenotypes, *pSUC2::amiRSUL^hst-1/-25/-26^* leaves exhibited higher *SUL*/SUL mRNA/protein levels than *pSUC2::amiRSUL* leaves (Fig 2D; Appendix Figs S8–S9). We first considered that the *hst* background might impair amiRSUL silencing by preventing amiRSUL accumulation. *pSUC2::amiRSUL^hst-1/-25/-26^* leaves showed a ~ 30% reduction in amiRSUL levels (Fig 3A). This difference was not due to a peculiarity of *hst* causing reduced *pSUC2* activity (Appendix Fig S9) but was in line, instead, with the originally reported effects of *hst-1* on the steady-state levels of some, yet not other, endo-miRNAs (Park *et al*, 2005). Accordingly, a recent sRNA sequencing analysis conducted in *Arabidopsis hst-15* plants revealed that ~ 30% of all endo-miRNAs show reduced levels in this background, albeit to greatly varying extents (Cambiagno *et al*,

2021) (Appendix Fig S10A). The levels of endo-miRNAs were also differently affected by *hst-1*, *hst-25*, and *hst-26* depending on the species under consideration; miR160, miR165, miR173, and miR822 accumulation was barely altered, if at all, or even slightly higher in certain *hst* alleles, compared to that of the more substantially reduced miR159, miR168, or miR171 (Fig 3A). Halving amiRSUL dosage in *pSUC2::amiRSUL^+/−^* hemizygous plants (Fig 3B, left panel) resulted in a slightly less extensive yet readily detectable amiRSUL-silencing phenotype resembling that of the *pSUC2::amiRSUL^+/+^* homozygous parental line. Quantifying the *SUL* mRNA levels in WT, *pSUC2::amiRSUL^+/−^* and *pSUC2::amiRSUL^+/+^* leaves confirmed these observations (Fig 3B, right panel). The slightly reduced movement phenotype of *pSUC2::amiRSUL^+/−^* plants was, however, incommensurate, in extent, with the barely detectable movement phenotypes displayed by *pSUC2::amiRSUL^+/+^* homozygous leaves with *hst* mutant backgrounds (Fig 3B compared to Fig 2C). To explain this apparent discrepancy, we first considered the possibility that the ~ 60% amiRSUL remaining in *pSUC2::amiRSUL^+/−^* plants with the WT background (Fig 3C) might be fully active whereas the ~ 70% remaining in *pSUC2::amiRSUL^+/+^* plants with the *hst* background (Fig 3A) might be sub-effective, for instance as a consequence of tailing/trimming/misprocessing, inefficient loading, and/or reduced silencing activity. To explore the possibility that elevated miRNA tailing/trimming/misprocessing might occur in *hst*, we mined the recent *hst-15* versus WT sRNA sequencing data from Cambiagno *et al* (2021). Tailing/trimming was assessed with a method similar to that described recently in Giudicatti *et al* (2021), for which publicly available sRNA sequencing data for the *hen1* and *hen1 heso1* mutants (Wang *et al*, 2018) were also used in parallel. While tailing/trimming was readily detected, as reported, in the latter two mutants, it was not observed in *hst-15* (Appendix Fig S10B) and nor was miRNA misprocessing, as assessed by testing the accumulation of 20–22-nt-long miRNA isoforms deviating by up to +/− 5 nucleotides from their cognate, annotated sequences (Iki *et al*, 2018) (Appendix Fig S10B).

We then explored the possibility that the *hst* background might impair the execution of amiRSUL silencing. We first tested whether *hst* prevents amiRSUL loading into AGO1, the main amiRSUL- and

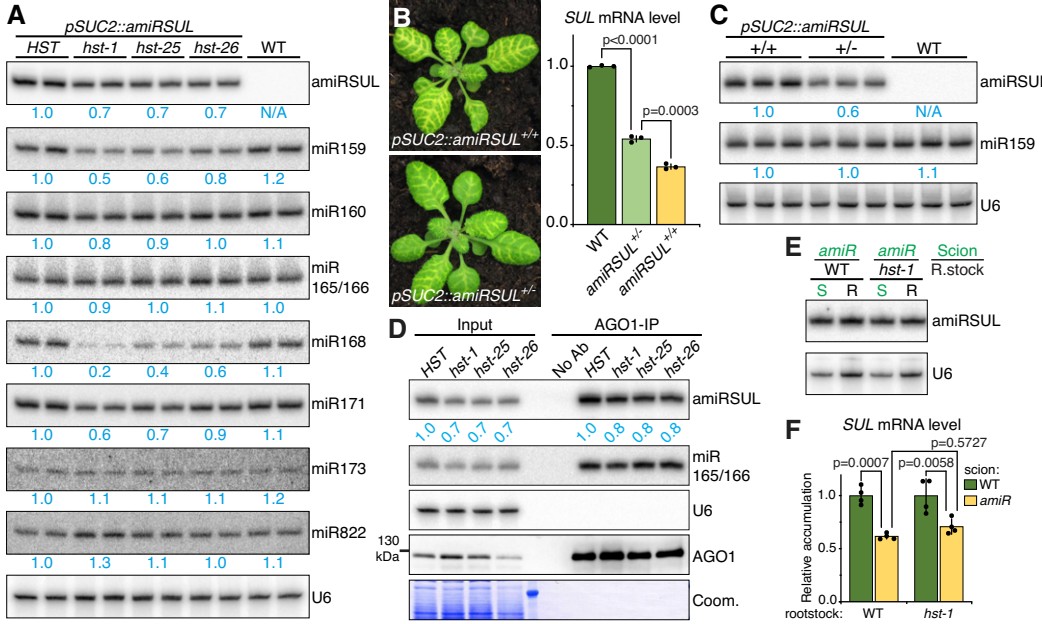

**Figure 3. amiRSUL levels are reduced in *hst*, unlike its loading into AGO1 or its silencing activity.**

A Northern analysis of indicated miRNAs, in biological duplicates, in leaves with the indicated genotypes. U6 was probed as endogenous control. Average relative U6-normalized band-intensity quantifications are indicated.

B Left: silencing phenotype of homozygous (+/+) versus hemizygous (+/−) *pSUC2::amiRSUL* plants. Right: RT–qPCR analysis of *SUL* mRNA levels in leaves of (+/+) or (+/−) *pSUC2::amiRSUL* versus WT *Arabidopsis*. Error bars: SD. *t*-test *P*-values are indicated. *n* = 3.

C amiRSUL northern analysis, in biological triplicates, in leaves of (B). miR159 and U6 were probed as endogenous controls. Band-intensity quantifications: as in (A).

D amiRSUL northern analysis in input and AGO1-immunoprecipitated (AGO1-IP) fractions isolated from leaves of *pSUC2::amiRSUL* plants in the specified genotypes. No Ab: No antibody added to the *HST* input extract (IP negative control). miR165/166 and U6 were probed as endogenous controls. AGO1 western analysis and Coomassie blue (Coom.) staining of the Western blot membrane are provided as controls. amiRSUL band-intensity quantifications, U6-normalized for input samples, are indicated.

E amiRSUL northern analysis in *pSUC2::amiRSUL* (*amiR*) scions (S) and non-transgenic WT and *hst-1* rootstocks (R) in the indicated grafting combinations. U6: as in (A).

F RT–qPCR analysis of *SUL* mRNA levels in WT and *hst-1* rootstocks grafted onto WT or *amiR* scion. Error bars: SD. Tukey's multiple comparisons test *P*-values are indicated. *n* = 4.

Source data are available online for this figure.

endo-miRNA-silencing effector protein. Neither the hypomorphic *hst-25/-26* nor the loss-of-function *hst-1* mutation overtly altered AGO1 protein accumulation as compared to that in leaves of the *pSUC2::amiRSUL* parental line (Fig 3D). Moreover, a similar proportion of AGO1-immunoprecipitated *versus* total amiRSUL was detected in the WT and the three *hst* mutant backgrounds (Fig 3D). Similar findings made with endo-miR165/166 suggested, therefore, that AGO1 loading efficacy is not overtly altered in either hypomorphic or loss-of-function *hst* mutants. To test, finally, whether the silencing capacity of the AGO1:amiRSUL complex *per se* remains functional in the *hst-1* null background, we grafted *pSUC2::amiRSUL* transgenic scions onto either non-transgenic WT or non-transgenic *hst-1* rootstocks. We found that amiRSUL was equally abundant, and *SUL* silencing equally effective, in both types of rootstocks (Fig 3E and F). This suggests, therefore, that AGO1 loaded with amiRSUL synthesized under WT processing conditions is equally efficient in mediating silencing in a *hst* as it is in a WT background. We note that 20 out of 249 (8%) miRNA-target transcripts, as collated in Arribas-Hernández *et al* (2016), show significantly increased levels in RNA sequencing analyses conducted in *hst-15* versus WT plants (Cambiagno *et al*, 2021) (Appendix Fig S10C), of which 8 (3%) are targets of miRNAs reported as being significantly

down-regulated (Cambiagno *et al*, 2021) (Appendix Fig S10A and D). Collectively, these results indicate that neither the processing, loading efficacy, nor the activity of amiRSUL is significantly impaired in *hst*. The reduction, by *hst*, in amiRSUL accumulation is consistent with an effect on its biogenesis as recently reported for ~ 30% of endo-miRNAs (Cambiagno *et al*, 2021). The above results indicate, however, that this reduction is unlikely to explain, alone, the strongly reduced amiRSUL-silencing phenotypes seen in *pSUC2::amiRSUL*[+/+] in the *hst* background (Fig 2C) compared with the modestly reduced phenotype seen in *pSUC2::amiRSUL*[+/−] in the WT background (Fig 3B). The last available hypothesis to explain this discrepancy is that, in addition to the biogenesis of amiRSUL, HST controls its movement. Given the ambiguity yielded here by an indirect assessment of amiRSUL mobility via its activity, we decided to test this hypothesis directly by comparing the physical movement of amiRSUL in the *hst* versus WT background.

## HST is required for amiRSUL movement

To quantify the physical movement of amiRSUL between cells, we used Meselect (Svozil *et al*, 2016; Brosnan *et al*, 2019)—which mechanically separates the vasculature from the epidermis—in

*pSUC2::amiRSUL* versus *pSUC2::amiRSUL^{hst-1/-25/-26}* leaves. In four independent biological replicates, successful separation was confirmed by the respective strong enrichments of the *SUC2* (leaf vasculature-specific) and *ATML1* (leaf epidermis-specific) transcripts (Appendix Fig S11). Phosphorimager-based signal quantification from northern analyses of the four replicates consistently revealed a reduction in amiRSUL levels in the *hst*-recipient epidermis significantly exceeding that in the *hst* emitting vasculature (Fig 4A; Appendix Fig S12A). This strongly suggests that, in addition to its biogenesis in amiRSUL-emitting cells, HST is indeed required for amiRSUL movement. Based on epidermis/vasculature ratio averages, we estimate that amiRSUL movement was reduced by, respectively, 59, 68, and 64% in *hst-1*, *hst-25*, and *hst-26* compared with WT background (Fig 4B). These results therefore uncover a potent, albeit possibly incomplete, effect of *hst* on amiRSUL movement.

Signal quantification from northern analyses of four independent grafting experiments also consistently revealed that the reduction in amiRSUL levels in recipient non-transgenic WT rootstocks exceeded substantially that observed in emitting *pSUC2::amiRSUL^{hst-1}* and *pSUC2::amiRSUL^{hst-25}* scions (Fig 4C, Appendix Fig S12B). Based on rootstock/scion ratio averages, we estimate that amiRSUL long-distance movement was reduced by, respectively, 69 and 53% in the *hst-1* and *hst-25* compared with WT background (Fig 4D). Importantly, two of the four grafting experiments included two well-characterized mutants known to strongly impact miRNA biogenesis either by impeding miRNA processing (*hyl1-2*) or by reducing miRNA stability post-processing (*hen1-6*) (Fig 4C). Northern analysis confirmed that amiRSUL levels were substantially lower in *pSUC2::amiRSUL^{hen1-6}* and *pSUC2::amiRSUL^{hyl1-2}* scions (71 and 68% reduction, respectively) than they were in *pSUC2::amiRSUL^{hst-1}* and *pSUC2::amiRSUL^{hst-25}* scions (44% and 23% reduction, respectively) as compared, in each case, to *pSUC2::amiRSUL* tissues (Fig 4C). This was fully consistent with the northern analyses of independent grafting experiments already presented, for distinct purposes, in Fig 1G for *pSUC2::amiRSUL^{hen1-14}* (70% amiRSUL reduction) and in Appendix Fig S3A for *pSUC2::amiRSUL^{hyl1-2}* (68% amiRSUL reduction). Signal quantifications in all these grafting experiments were used to generate the rootstock/scion ratio averages presented in Fig 4D, which revealed that movement of the residual amiRSUL derived from *pSUC2::amiRSUL^{hen1-6/hen1-14}* or *pSUC2::amiRSUL^{hyl1-2}* scions was not significantly reduced (*hen1* backgrounds) or reduced by only 20% (*hyl1-2* background), as compared to the WT background. This was in stark contrast with

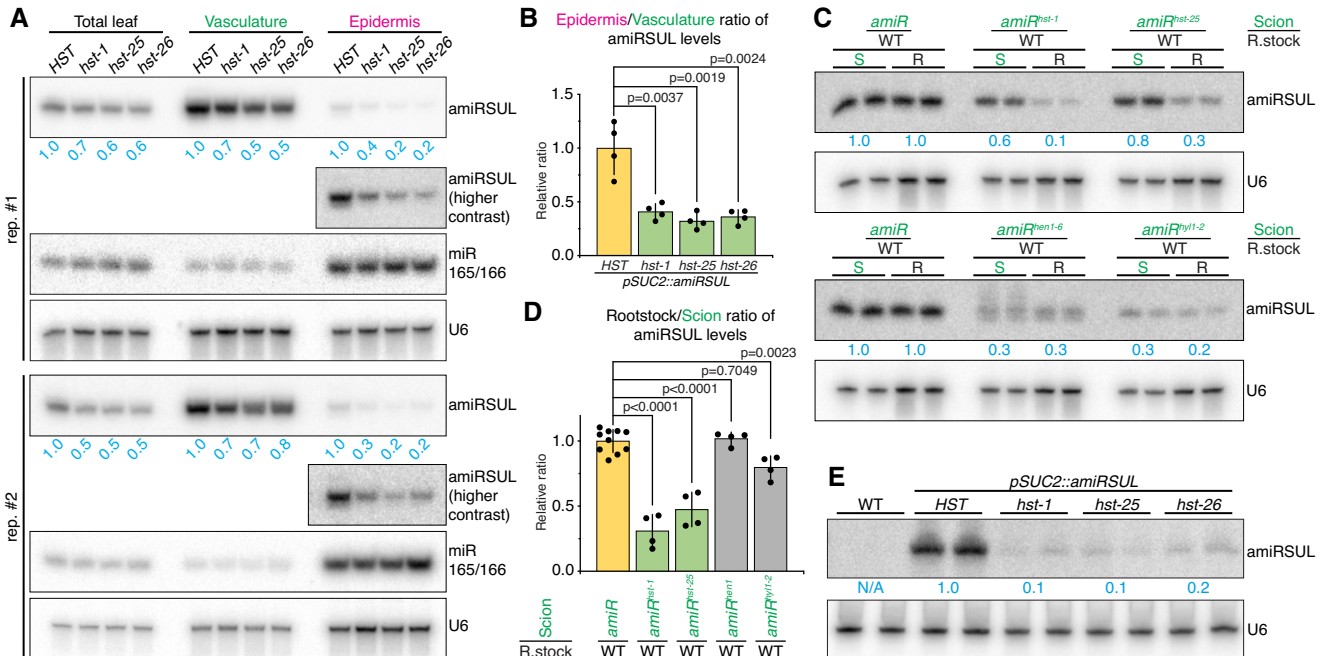

**Figure 4.  HST is required for amiRSUL cell-to-cell movement and long-distance movement.**

A   amiRSUL northern analysis, in two independent replicates (rep.), in Meselect-separated leaf vasculature versus epidermis of *pSUC2::amiRSUL* plants with the specified genotypes (see also Appendix Fig S12A). An enhanced contrast is shown for the epidermis. miR165/166 and U6 were probed as endogenous controls. U6-normalized band-intensity quantifications relative to *HST* are indicated for each tissue.

B   Average epidermis/vasculature ratio of amiRSUL levels presented in (A) and Appendix Fig S12A. Error bars: SD. *t*-test *P*-values are indicated. *n* = 4.

C   amiRSUL northern analysis, in biological duplicates, in scions (S) and rootstocks (R) in the specified genotypes' grafting combinations (*amiR*: *pSUC2::amiRSUL*) (see also Appendix Fig S12B). U6: as in (A). Average relative U6-normalized band-intensity quantifications are indicated for each tissue.

D   Average rootstock/scion ratio of amiRSUL levels as measured in (C), Fig 1G, Appendix Figs S3A and S12B. Error bars: SD. *t*-test *P*-values are indicated. *n* ≥ 4.

E   amiRSUL northern analysis, in biological duplicates, from total RNA extracted from aphids fed on WT or *amiR* plants, in the specified genetic backgrounds. U6: as in (A). Average relative U6-normalized band-intensity quantifications are indicated.

Source data are available online for this figure.

the 69% and 53% reductions in movement observed with the *hst-1* and *hst-25* scions, respectively (Fig 4D). These results indicate that decreasing amiRSUL biogenesis in silencing-emitting tissues—even very substantially (up to 71%)—does not suffice, *per se*, to cause a strong deficit in movement efficacy, further emphasizing the singular effect of *hst* on this process.

Consistent with *hst* compromising amiRSUL access to the phloem translocation stream within the SEs, aphids fed on *pSUC2::amiR-SUL^hst-1/-25/-26* plants barely accumulated amiRSUL compared with those fed on *pSUC2::amiRSUL* plants (Fig 4E). The *hst* background unlikely impacted the aphids' feeding behavior because aphids displayed unaltered survival rates in previous artificial diet experiments conducted on several miRNA-deficient mutant compared with WT *Arabidopsis* (Kettles *et al*, 2013). Moreover, no overt changes were observed in the numbers of aphids populating the WT as opposed to *hst* mutant plants in the short time-frame of our feeding experiments whose analyses involved the same quantities of aphids isolated from each genotype. Collectively, the results of the Meselect and grafting experiments support the notion that, in addition to modulating amiRSUL biogenesis in emitting cells, HST is required for its movement between cells and across organs via the phloem.

## HST is required in amiRSUL-emitting but not amiRSUL-recipient tissues

While the effects of HST on amiRSUL biogenesis must be *de facto* restricted to the silencing-emitting CCs, its additional effects in controlling amiRSUL movement might require its presence in silencing-emitting cells, silencing-receiving cells, or a combination of both. To explore more precisely where, in space, this control might be exerted, we conducted mosaic rescue experiments by transforming *pSUC2::amiRSUL^hst-1* loss-of-function plants with the *pSUC2::HST:GFP* transgene, having validated the functionality of HST:GFP in the genetic complementation assays presented in Appendix Fig S6B. *pSUC2::HST:GFP* restricts HST:GFP expression exclusively to the amiRSUL-emitting CCs, as opposed to the amiRSUL-receiving cells. These cells must at least include the mesophyll cells which, unlike epidermal cells, are photosynthetically active and, hence, prone to chlorosis caused by *SUL* silencing. We found that *pSUC2::HST:GFP* rescued the vein-centered chlorosis as well as the aberrant leaf shape, but not the defective phyllotaxis of *hst-1* (Fig 5A; Appendix Fig S13A and B). These effects were unlikely caused by spurious or position-dependent ectopic expression of *pSUC2::HST:GFP* because they were identically observed in 28 out of 28 independent transformants. All exhibited a vein-centered chlorotic pattern similar, in extent and intensity, to that seen in leaves of the parental *pSUC2::amiRSUL* line (Fig 5A; Appendix Fig S13A). These results indicate that selectively restricting HST's expression into the amiRSUL-emitting CCs, in the *hst-1* loss-of-function background, suffices to recapitulate amiRSUL activity in at least the neighboring mesophyll cells. We could not conduct, in *pSUC2::amiRSUL^hst-1* leaves, the converse mosaic rescue experiment in which HST:GFP expression would be restricted to amiRSUL-recipient mesophyll cells, as opposed to amiRSUL-emitting CCs. Indeed, we could not identify, in the *Arabidopsis* literature, a strict mesophyll-specific promoter notably devoid of any additional vascular activity. Nonetheless, the fact that *SUL* silencing was equally effective in both non-transgenic WT and non-transgenic *hst-1* rootstocks grafted onto *pSUC2::amiRSUL* transgenic scions (Fig 3F)

indicates that HST's function is dispensable for silencing movement and execution in amiRSUL-recipient cells, at least in the grafting setting. These collective analyses of cell-to-cell and long-distance movement therefore suggest that HST is required for amiRSUL emission from incipient cells, but not reception or activity, in recipient cells.

## HST is required cell autonomously for amiRSUL movement

While amiRSUL-silencing execution unlikely requires HST:GFP expression in amiRSUL-recipient cells, the rescue of amiRSUL movement by *pSUC2::HST:GFP* might entail the physical translocation of HST:GFP from the CCs to at least the neighboring mesophyll cells. Alternatively, HST:GFP might act cell autonomously by promoting amiRSUL movement specifically and exclusively within the amiRSUL-emitting CCs. To distinguish between these possibilities, we explored the spatial distribution of HST:GFP in leaves and roots of *pSUC2::amiRSUL^hst-1* plants co-expressing *pSUC2::HST:GFP*. As controls for those experiments, *pSUC2::GFP* and *pSUC2::tmGFP9* transgenic plants were grown in parallel, under the same conditions. *pSUC2::GFP* was previously used to document the extent of free GFP movement from the CCs to neighboring cells and over long distance *via* the SEs (Imlau *et al*, 1999). *pSUC2::tmGFP9* was also previously used to provide a reference signal from a membrane-anchored, i.e., non-mobile, GFP allele (Stadler *et al*, 2005). In macroscopic observations conducted under UV illumination of whole seedlings, the signal from *pSUC2::GFP* was widespread throughout the lamina of young emerging leaves (Fig 5B), demonstrating the phloem unloading and cell-to-cell movement of free GFP in these sink tissues, as previously reported (Imlau *et al*, 1999). Also as previously reported (Stadler *et al*, 2005), the signal from *pSUC2::tmGFP9* remained, by contrast, restricted to the vasculature in equivalent leaves (Fig 5B). The signal from *pSUC2::HST:GFP* was, however, below the detection limit of macroscopic assessment, prompting us to use confocal microscopy instead. As was observed, as expected, with the *pSUC2::tmGFP9* reference signal, we found that the *pSUC2::HST:GFP* signal was exclusively CC-restricted in young emerging leaves (Fig 5C), coinciding with the rescue of amiRSUL movement and activity (Fig 5A). As reported (Truernit, 2019), the CC-restricted tmGFP9 signal was manifested as cytosolic round clusters corresponding to membrane aggregates. That of HST:GFP was concentrated in the typically elongated nuclei of CCs and was also visible, albeit to a lower extent, in their cytosol, consistent with the proposed XPO5 function of HST. Similar observations were made in mature roots (Fig 5D), making it unlikely, therefore, that HST:GFP, in contrast to amiRSUL, moves from CCs to neighboring cells.

To investigate whether HST:GFP expressed from *pSUC2::HST:GFP* might move over long distances via the phloem SEs, we inspected root tips in which the meristem-proximal division zone is a major phloem unloading domain for diverse macromolecules including proteins and RNA (Ross-Elliott *et al*, 2017). Phloem unloading and subsequent cell-to-cell movement of free GFP were readily detected, under confocal microscope, throughout the division zone of *pSUC2::GFP* root tips (Fig 5E). In sharp contrast, the tmGFP9 signal from *pSUC2::tmGFP9* remained confined within the CCs of the meristem-distal differentiation zone, agreeing with the cognate expression pattern of the *pSUC2* promoter previously documented in *Arabidopsis* root tips (Stadler *et al*, 2005). Likewise, the

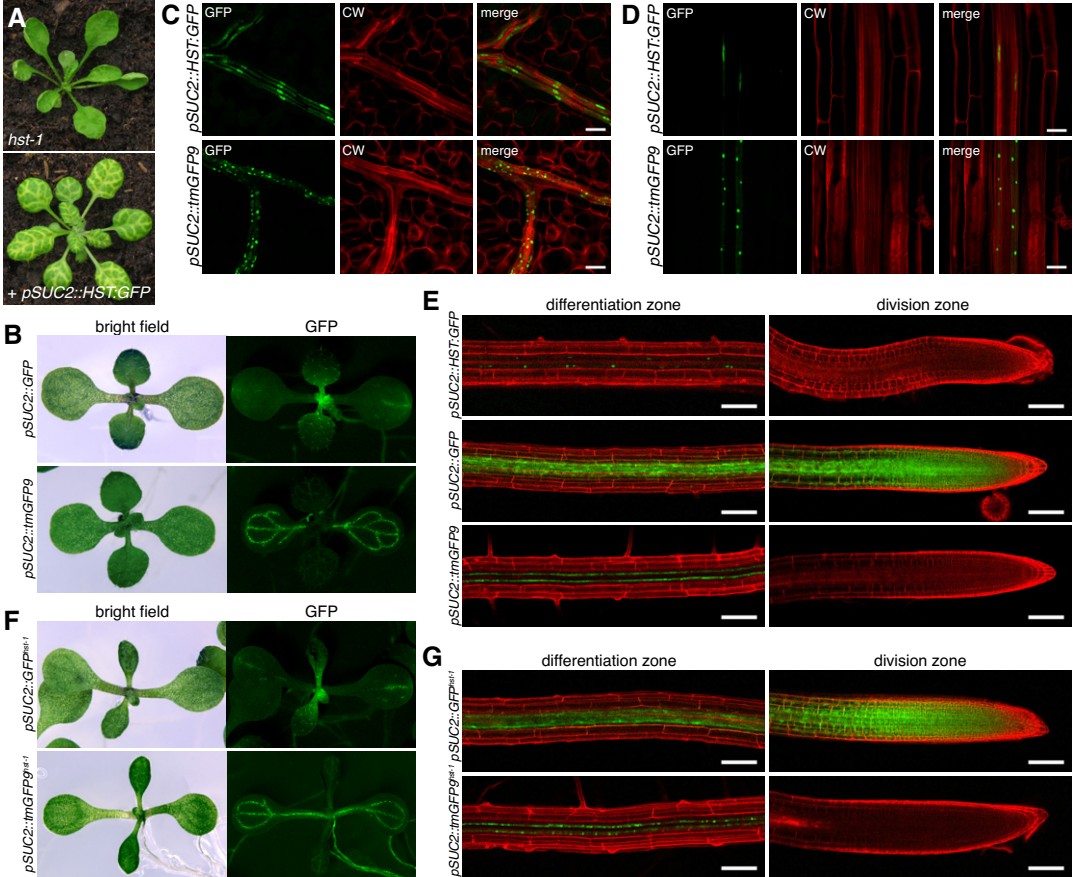

**Figure 5.  A cell-autonomous activity of HST enables movement of amiRSUL, but not of free GFP.**

A   Representative vein-chlorosis phenotype caused by the genetic mosaic rescue of *pSUC2::HST:GFP* co-expressed in *pSUC2::amiRSUL*[hst-1] plants (see also Appendix Fig S11A).

B   Bright field and GFP fluorescence images of *pSUC2::GFP* and *pSUC2::tmGFP9* seedlings.

C, D  Localization of HST:GFP and, as a reference, of tmGFP9 expressed under the *pSUC2* promoter in *Arabidopsis* primary leaves (C) or roots (D). CW: Calcofluor White staining. Scale bars: 20 µm.

E   Confocal fluorescence images of the differentiation and division zones of *Arabidopsis* root tips expressing *pSUC2::HST:GFP*, *pSUC2::GFP* or *pSUC2::tmGFP9*. Scale bars: 100 µm.

F   Bright field and GFP fluorescence images of *pSUC2::GFP*[hst-1] and *pSUC2::tmGFP9*[hst-1] seedlings.

G   Confocal fluorescence images of the differentiation and division zones of *pSUC2::GFP*[hst-1] or *pSUC2::tmGFP9*[hst-1] root tips. Scale bars: 100 µm.

Source data are available online for this figure.

signal from HST:GFP expressed from *pSUC2::HST:GFP* was restricted to the CCs in the differentiation zone without any detectable sign of phloem unloading in the downstream division zone (Fig 5E). This indicates that HST:GFP, in contrast to free GFP, unlikely moves over long distances within the SEs. Collectively, these results suggest that the rescue, by *pSUC2::HST:GFP*, of amiRSUL movement in the *hst-1* background entails a cell-autonomous action of HST:GFP. By extension, HST's requirement as a facilitator of amiRSUL movement in *pSUC2::amiRSUL* plants is likely circumscribed to the amiRSUL-emitting CCs as opposed to amiRSUL-recipient cells.

**Phloem unloading and cell-to-cell movement of free GFP are not compromised in the *hst-1* background**

The effect of *hst* on amiRSUL emission might entail a generic hindrance to macromolecular transport, which would be manifested

as impaired phloem unloading and cell-to-cell movement of free GFP expressed from *pSUC2::GFP*. To test this idea, the *hst-1* loss-of-function mutation was introgressed into *pSUC2::GFP*- and, as a negative control, *pSUC2::tmGFP9*-transgenic plants. In macroscopic observations conducted under UV illumination of whole seedlings, the phloem unloading and subsequent cell-to-cell movement of free GFP expressed from *pSUC2::GFP*[hst-1] were as extensive in the lamina of young emerging leaves as they were in equivalent tissues of *pSUC2::GFP* plants (Fig 5F compared to Fig 5B). The vasculature-restricted signal from tmGFP9 also remained unaltered in equivalent leaves of *pSUC2::tmGFP9* versus *pSUC2::tmGFP9*[hst-1] transgenic plants (Fig 5F compared to Fig 5B). In confocal microscopy analyses, free GFP was equally phloem-unloaded and as widely distributed in the division zone of *pSUC2::GFP*[hst-1] as it was in that of *pSUC2::GFP*- root tips (Fig 5G). The tmGFP9 signal, by contrast, remained CC-restricted in the meristem-distal differentiation zones

of both *pSUC2::tmGFP9* and *pSUC2::tmGFP9^hst-1^* root tips (Fig 5G). Therefore, the *hst-1* mutant background is unlikely to impact amiRSUL movement by impairing macromolecular trafficking from CCs to neighboring cells, or over long distances via phloem-based translocation and unloading.

### HST nucleo-cytosolic shuttling is not required for amiRSUL movement

We found that HST:GFP constitutively expressed from the ubiquitin 10 promoter (*pUBQ::HST:GFP*) fully rescues amiRSUL movement in *pSUC2::amiRSUL^hst-1/-25/-26^* plants (Appendix Fig S6B). In agreement with recent work (Cambiagno *et al*, 2021), confocal microscopy analyses conducted in the root tips of these plants consistently revealed a dominant nucleoplasmic and fainter cytosolic localization for HST:GFP (Fig 6A). This confirmed the observations already made with the CC-restricted signals in leaves and roots of *pSUC2::amiRSUL^hst-1^* plants co-expressing *pSUC2::HST:GFP* (Fig 5C and D). The subcellular distribution of HST is therefore consistent with the nucleo-cytosolic shuttling of an XPO5 ortholog as originally proposed (Bollman *et al*, 2003). A role for HST in controlling macromolecular trafficking being unlikely, we asked if its requirement for amiRSUL movement entailed nucleocytoplasmic shuttling. We used the *hst-3* mutant allele, which displays similar developmental defects (Telfer & Poethig, 1998) and the same reduced accumulation of some, but not other, miRNAs as observed in the reference loss-of-function *hst-1* mutant (Park *et al*, 2005). In *hst-3*, a 3 bp deletion located in exon 1 (N-terminal domain) replaces amino acids $D^{36}S^{37}$ with an alanine. In the yeast two-hybrid assay, this reduces the mutant hst-3 protein's interaction with *Arabidopsis* RAN1 (Bollman *et al*, 2003). RAN1 is generally considered mandatory for nuclear XPO5:cargo interaction and subsequent cytosolic cargo-release, in a GTP-dependent manner (Cautain *et al*, 2015). Accordingly, recent work shows that HST's nucleo-cytosolic shuttling is compromised in the *Arabidopsis ran1* mutant (Cambiagno *et al*, 2021).

Introgressing *hst-3* into the *pSUC2::amiRSUL* background resulted in strongly reduced vein-centered chlorosis (Fig 6B). Conversely, *pUBQ::hst-3:GFP* barely rescued the amiRSUL movement phenotype when introduced into the *pSUC2::amiRSUL^hst-1^* background (Appendix Fig S14). This was despite the accumulation of the hst-3:GFP mutant protein being similar to that of HST:GFP produced from *pUBQ::GFP* in *pSUC2::amiRSUL^hst-1^* plants displaying, by contrast, full rescue of amiRSUL movement (Appendix Fig S14). Unlike the nuclear retention expectedly caused by the *hst-3* lesion, however, the subcellular distribution of hst-3:GFP was not overtly changed compared with that of HST:GFP (Fig 6C); it was still mostly nuclear with a fainter yet readily detectable cytosolic signal. Thus, while it compromises HST:RAN1 interaction *in vitro*, the punctual $D^{36}S^{37}$->A mutation does not overtly affect nucleo-cytosolic shuttling of hst-3 *in vivo*, presumably because other signals located in HST's N-terminal domain contribute to this process. Recent work shows that a 107 amino acid deletion indeed causes cytosolic retention of the resulting N-truncated protein (HST$^{\Delta N}$), which, in turn, compromises the nuclear biogenesis, but not nuclear export, of miRNAs (Cambiagno *et al*, 2021). The results obtained here with the *hst-3* mutant and ectopic hst-3:GFP protein expression make it unlikely, therefore, that HST's nucleo-cytosolic shuttling is required for the control of amiRSUL movement.

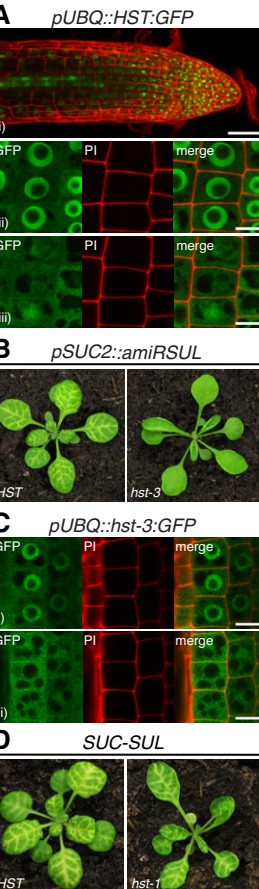

**Figure 6. HST's nucleo-cytosolic shuttling is dispensable for its control of amiRSUL movement.**

A  Subcellular localizations of HST:GFP expressed from the *UBQ10* promoter (*pUBQ*) in *Arabidopsis* roots (i). Distinct planes of the same root cells imaged by confocal microscopy are shown (ii–iii). PI: propidium iodide staining. Scale bars: 50 μm (i); 10 μm (ii–iii).

B  Phenotype of *pSUC2::amiRSUL* in the *HST* or *hst-3* background.

C  Subcellular localizations of hst-3:GFP expressed from *pUBQ* in root cells. Distinct planes of the same root cells imaged by confocal microscopy are shown (i–ii). PI: as in (A). Scale bars: 10μm.

D  Phenotype of *SUC-SUL* plants expressing, from the *pSUC2* promoter, an inverted-repeat transgene producing siRNA populations targeted against *SUL*, in the *HST* or *hst-1* background.

Source data are available online for this figure.

### *hst-1* does not suppress mobile *SUL* silencing triggered by the *SUC-SUL* inverted-repeat transgene

It was previously noted that *hst-1* loss-of-function plants display unaltered steady-state accumulation of several endogenous siRNA species, contrasting with the reduced levels observed for some, albeit not other, endo-miRNAs (Park *et al*, 2005). It was also noted, in the same study, that *hst-1* does not suppress sense-PTGS spontaneously initiated in *Arabidopsis* by the L1 locus, which contains a *35S::GUS* transgene. Of significance here, it was later found that, upon its sporadic initiation probably in only a few cells, L1 silencing invades the remaining plant tissues by virtue of amplification and movement of GUS-derived siRNAs; accordingly, L1 silencing is

efficiently graft-transmitted to non-silenced GUS transgenic tissues (Taochy *et al*, 2019). Thus, the fact that S-PTGS of L1 is unaltered in the *hst-1* background suggested that HST might not be required for transgene-derived siRNA movement. To ascertain this notion unambiguously and under *pSUC2::amiRSUL*-comparable conditions, we generated *SUC-SUL^{hst-1}* transgenic plants by introgressing *hst-1* into the *pSUC2*-driven *SUL*-derived *IR* transgene (*SUC-SUL*) system (Himber *et al*, 2003). In *SUC-SUL* plants, vein-centered chlorosis akin to that observed in *pSUC2::amiRSUL* plants is achieved by movement of processed siRNAs, 21-nt and 24-nt in length, that are spawned as a large population from a *SUL*-derived long dsRNA expressed solely within the CCs under *pSUC2* (Devers *et al*, 2020). Of the 21-nt and 24-nt siRNA species, only the former are required for mobile silencing, which, like amiRSUL silencing, is executed in recipient cells in a prevailing AGO1-dependent manner (Jay *et al*, 2019; Devers *et al*, 2020). As shown in Fig 6D, vein-centered chlorosis was as extensive in *SUC-SUL^{hst-1}* as it was in *SUC-SUL* leaves, indicating that HST is neither required for the production, nor for the mobility or activity of 21-nt *SUL*-derived siRNAs.

### *hst-1* suppresses movement of endo-miRNAs displaying otherwise negligible or no impairment in their steady-state accumulation

The results obtained with the artificial amiRSUL system prompted us to investigate whether HST is required for functional endo-miRNA movement. To address this issue, we examined the effects of the *hst-1* loss-of-function mutation on the steady-state accumulation and physical movement of the stress-induced miR395 and the constitutively expressed miR160. Both display well-documented non-cell-autonomous activities amenable to exploration via grafting and/or Meselect (Brosnan *et al*, 2019), as used here with amiRSUL. miR395 is induced in shoots of plants subjected to $SO_4$ starvation and accumulates in *Arabidopsis* roots in a graft-transmissible manner (Kawashima *et al*, 2009; Buhtz *et al*, 2010). As for amiRSUL (Fig 1E and F), this presumably involves the translocation of processed miR395 via the phloem, its demonstrated cell-to-cell movement form (Brosnan *et al*, 2019). Consistent with this idea, miR395 accumulated in miRNA-processing-deficient *hyl1-2* recipient rootstocks grafted onto $SO_4$-starved WT scions. In two independent northern analyses of four biological replicates, the steady-state accumulation ratio of miR395 in $SO_4$-starved *hst-1 versus* WT scions was 0.9 and 1.1, compared to 0.4 and 0.5 in $SO_4$-starved *hyl1-2 versus* WT scions (Fig 7A). Thus, unlike the miRNA-processing-deficient *hyl1-2* background, the *hst-1* background incurred only a marginal decrease or even a slight increase to miR395 levels in miR395-emitting scions. This result is consistent with *hst-1* affecting the biogenesis of some, but not other, miRNAs in leaves (Fig 3A) and with the levels of miR395 not being significantly changed in the sRNA sequencing data available for *hst-15 versus* WT *Arabidopsis* (Cambiagno *et al*, 2021) (Appendix Fig S10A). In contrast to the negligible effects of *hst-1* in scions, amiRSUL accumulation was significantly reduced in the corresponding recipient rootstocks, as assessed by signal quantification (Fig 7A). Relative quantification, by stem-loop RT–qPCR, of miR395 levels in scions and rootstocks from the four replicates confirmed these observations. Based on the ensuing rootstock/scion ratio averages, we estimate that miR395 movement in recipient *hyl1-2* rootstocks was reduced by 54% in

grafts involving *hst-1-* compared with grafts involving WT scions (Fig 7B, upper panel). This figure was in line with the long-distance movement of amiRSUL being decreased, on average, by 61% in the *hst-1* and *hst-25* versus WT backgrounds (Fig 4C and D, Appendix Fig S12B). These findings are of functional significance because the levels of *APS4* transcripts, targeted by miR395, were increased by 78% in recipient *hyl1-2* rootstocks in grafts involving *hst-1-* compared with WT miR395-emitting scions (Fig 7B, lower panel). These results suggest that HST facilitates functional miR395 long-distance movement with negligible or no effect on its steady-state accumulation within miR395-emitting tissues.

Analyses in aerial, as opposed to below-ground tissues, involved miR160, which accumulates as a processed species in the leaf epidermis despite a strictly vasculature-restricted transcription pattern, providing an indication of its movement (Brosnan *et al*, 2019). As with amiRSUL, we used Meselect to successfully separate the vasculature from the epidermis in leaves of WT as opposed to *hst-1* loss-of-function plants (Appendix Fig S15). Signal quantification from northern analyses of three independent replicates revealed that the steady-state accumulation ratio of miR160 in the *hst-1 versus* WT vasculature was 1.0, 0.8, and 0.8. Thus, the *hst-1* background incurred only marginal, if any, change to miR160 levels in miR160-emitting vascular cells (Fig 7C). This was consistent with independent northern analyses also conducted in leaves of *hst-1*, *hst-25*, and *hst-26* (Fig 3A) and with the levels of miR160 not being significantly changed in the sRNA sequencing data available for *hst-15 versus* WT *Arabidopsis* (Cambiagno *et al*, 2021) (Appendix Fig S10A). Strikingly, however, the reduction in miR160 levels in the recipient epidermis significantly exceeded that in the emitting vasculature (Fig 7C). Relative quantification, by stem-loop RT–qPCR, of miR160 levels in the three replicates confirmed these observations. Based on the ensuing epidermis/vasculature ratio averages, we estimate that the vasculature-to-epidermis movement of miR160 was reduced by 63% in *hst-1*, compared with WT leaves (Fig 7D, upper panel). This was in line with the cell-to-cell movement of amiRSUL being decreased, on average, by 64% in the *hst-1*, *hst-25*, *and hst-26* compared with WT epidermis (Fig 4A). As for miR395, these findings are of functional significance because the levels of *ARF17* transcripts, targeted by miR160, accumulated ~5 -times more in *hst-1 versus* WT epidermal but not vascular cells (Fig 7D, lower panel). These results suggest that HST facilitates functional miR160 movement with negligible or no effect on its steady-state accumulation within miR160-emitting tissues. Collectively, the results support our findings with amiRSUL by uncovering a role for HST in facilitating both long-distance (miR395) and cell-to-cell (miR160) movement of endo-miRNAs.

### HST is required for cognate xylem pole specification mediated by the non-cell autonomous action of miR165/166 in the root

Our findings with miR160 and miR395 prompted us to revisit seminal work conducted on the *Arabidopsis* root's xylem cell development (Carlsbecker *et al*, 2010). Cognate specification of two outer protoxylem and two inner metaxylem files in the stele's xylem pole entails a decreasing RNA-turnover gradient of *HD-ZIP III* transcription factors (TFs) spanning the stele's outer-to-inner layers (Carlsbecker *et al*, 2010). This process is compromised in roots expressing miR165/166-resistant alleles of *HD-ZIP III* or in roots carrying

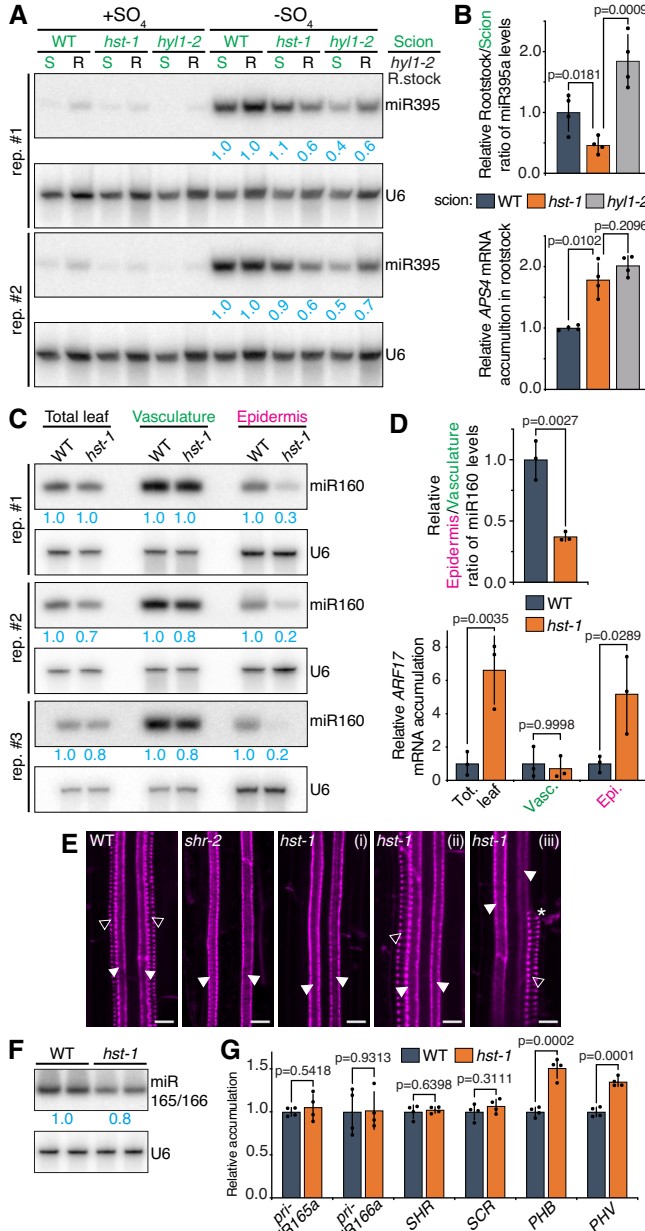

**Figure 7.** *hst-1* suppresses movement of endogenous miR395 and miR160 and alters xylem pole specification by mobile miR165/166.

A   miR395 northern analysis in scions (S) and *hyl1-2* rootstocks (R), in the indicated genotypes' combinations, under SO4-sufficient (+SO4) or SO4-starved (-SO4) conditions, in two biological replicates (rep.). U6 was probed as an endogenous control. Relative U6-normalized band-intensity quantifications are indicated for each tissue in -SO4 conditions.

B   Stem-loop RT–qPCR-based analysis of rootstock/scion ratio of miR395a levels in plants grafted with *hyl1-2* rootstocks onto scions of indicated genotypes (top) and RT–qPCR analysis of APS4$^{miR395}$ mRNA levels in the corresponding *hyl1-2* rootstocks (bottom), under -SO4 conditions. Error bars: SD. *t*-test (top) and Welch's *t*-test (bottom) *P*-values are indicated. n = 4.

C   miR160 northern analysis in Meselect-separated leaf vasculature and epidermis in WT versus *hst-1* plants, in three biological replicates (rep.). U6: as in (A). Relative U6-normalized band-intensity quantifications are indicated.

D   Stem-loop RT–qPCR-based analysis of epidermis/vasculature ratio of miR160 levels (top) and RT–qPCR analysis of ARF17$^{miR160}$ mRNA levels (bottom) in the samples in (C). Error bars: SD. *t*-test (top) and Tukey's multiple comparisons test (bottom) *P*-values are indicated. n = 3.

E   Basic fuchsin staining of protoxylem (unfilled arrowheads) and metaxylem (filled arrowheads) in WT, *shr-2* or *hst-1* roots. *protoxylem gap. Scale bars: 10 μm.

F   miR165/166 northern analysis in WT versus *hst-1* roots. U6: as in (A). Average relative U6-normalized band-intensity quantifications are indicated.

G   RT–qPCR analysis of *pri-miR165a/-miR166a*, *SHR*, *SCR*, PHB$^{miR165/166}$, and PHV$^{miR165/166}$ mRNA levels in WT versus *hst-1* roots. Error bars: SD. *t*-test *P*-values are indicated. n = 4.

Source data are available online for this figure.

**Figure 7.**

initially proposed mobility-based gradient has proven difficult to ascertain experimentally, especially under natural miR165/166 expression conditions (Vaten *et al*, 2011; Fan *et al*, 2021). Under the above premises, our findings predicted that *hst* would likely impair miR165/166 endodermis->stele non-cell autonomous activity and would thereby display similar xylem cell specification defects to *shr*. Indeed, 12 out of 24 inspected *hst-1* roots displayed xylem defects indistinguishable from those of *shr-2* roots (Fig 7E(i)); the other half exhibited only one instead of two protoxylem files occasionally interrupted by sporadic gaps (Fig 7E(ii-iii)). As observed in leaves (~ 10% or no reduction; Fig 3A) and consistent with the levels of miR165/166 not being significantly changed in the sRNA sequencing data available for *hst-15* versus WT *Arabidopsis* (Cambiagno *et al*, 2021) (Appendix Fig S10A), miR165/166 accumulation was reduced by at most ~ 20% in *hst-1* compared with WT roots (Fig 7F). Those of *pri-miR165a/166a*, *SHR*, and *SCR* remained unchanged, yet *PHABULOSA* (*PHB*) and *PHAVOLUTA* (*PHV*) HD-ZIP III target accumulation was higher in *hst-1* roots compared with WT roots (Fig 7G). These results therefore support the idea that HST facilitates the non-cell autonomous action of miR165/166 required for cognate xylem pole specification in *Arabidopsis* roots (Carlsbecker *et al*, 2010).

# Discussion

## A previously unrecognized role for HST as a facilitator of miRNA movement

The *hst* mutation was isolated 23 years ago in a screen for juvenile-to-adult-phase transition defects (Telfer & Poethig, 1998),

mutations in *SHR* (e.g., *shr-2* mutant; Fig 7E), which, together with *SCARECROW* (*SCR*), activates miR165/166 transcription. The ensuing ectopic *HD-ZIP III* accumulation in the stele causes both types of mutant roots to lack one, or usually the two, protoxylem files with intact, or sometimes ectopic, metaxylem files (Carlsbecker *et al*, 2010). Because *MIR165/166* is exclusively transcribed, by SHR and SCR, in the endodermis located outside the stele, it was concluded that xylem cells are most likely specified via a miR165/166-concentration gradient established from the outer-to-inner stele (Carlsbecker *et al*, 2010). The likely symplastic mobility of miR165/166 or precursor(s) thereof during this process (Vaten *et al*, 2011; Fan *et al*, 2021) is supported by the xylem cell specification defects and ectopic *HD-ZIP III* phenotype exhibited by *Arabidopsis* deficient in the PD-associated RLKs, BAM1/2 (Fan *et al*, 2021). However, the

whereupon HST's subsequent identification as the *Arabidopsis* XPO5 ortholog suggested its possible involvement in nucleocyto-plasmic transport of molecules controlling morphogenetic pathways (Bollman *et al*, 2003). Morphogens typically act to create gene expression gradients within, as well as between, cells, and recent work strongly suggests that at least some plant miRNAs display the latter property (Skopelitis *et al*, 2017; Brosnan *et al*, 2019). Notably, miR390 alone strongly influences leaf polarity during primordium development (Chitwood *et al*, 2009) and indeed abnormal adaxial-ization is a key underpinning of the aberrant leaf shape originally reported in *hst* mutant plants (Telfer & Poethig, 1998; Bollman *et al*, 2003). The processed miR390 accumulates uniformly throughout young leaf primordia in which *MIR390* transcription, by contrast, is spatially restricted to the vasculature (Chitwood *et al*, 2009). Put together, these available observations suggested early on that miR390 emission from vascular cells and its reception in surround-ing cells are integral to a leaf developmental process that is perturbed in the *hst* background.

We confirmed, here, observations made in past and recent stud-ies that *hst* appears to modulate miRNA biogenesis but in a manner only clearly apparent for some, unlike other, miRNAs. In *hst-15* versus WT *Arabidopsis*, for instance, ~ 30% of all endo-miRNAs are significantly down-regulated, albeit to varying extents (Appendix Fig S10A) (Cambiagno *et al*, 2021). Likewise, an effect of *hst* was clearly manifested here on the steady-state accumulation of amiRSUL and other endo-miRNAs, whereas it was barely observed, if at all, with mobile endo-miR160, endo-miR395, and endo-miR165, consistent with sRNA sequencing analyses in *hst-15* versus WT *Arabidopsis* (Appendix Fig S10A) (Cambiagno *et al*, 2021). In all cases, however, the reduction in amiRSUL, miR160, and miR395 levels observed in miRNA-recipient tissues exceeded substantially that observed in miRNA-emitting tissues in both leaf-peeling and grafting experiments. Moreover, strongly reducing amiRSUL processing or stability, using *hyl1-2* or *hen1* mutations, was not suf-ficient, *per se*, to substantially compromise movement of the resid-ual amiRSUL accumulating in *pSUC2::amiRSUL^hen1* and *pSUC2:: amiRSUL^hyl1-2* scions. Therefore, unlike other miRNA biogenesis factors such as HYL1 or HEN1, HST appears to control the move-ment, in addition to the biogenesis, of miRNAs. This *hitherto* unknown property is in line with the general prediction of Bollman *et al* (2003) that HST controls one or several morphogenetic path-ways. However, our results indicate that its nucleo-cytosolic shut-tling—a recently confirmed property of HST (Cambiagno *et al*, 2021)—is unlikely to be required for miRNA movement, and neither is it required for miRNA nuclear export as shown in multiple studies (Bologna *et al*, 2018; Zhu *et al*, 2019; Zhang *et al*, 2020; Cambiagno *et al*, 2021).

While a case can be made for HST's involvement in enabling miRNA mobility, we note that neither inter-cellular nor phloem-based movement was completely suppressed by the loss-of-function *hst-1* mutation. This hints at possible functional redundancy within the 17 family members of HST paralogs in *Arabidopsis* (Bollman *et al*, 2003) or at the involvement of multiple parallel pathways for miRNA movement of which only some involve HST. Also poten-tially contributing to the incomplete effect of *hst*, longer RNA species (e.g., pri/pre-miRNAs) might partially rescue movement in the *hst* as opposed to WT background, as was suggested for pri/pre-miR163 movement under artificial miRNA-processing-deficient

conditions (Brosnan *et al*, 2019). The *pSUC2::amiRSUL* system inherently confers CCs a nexus role in enabling both types of amiRSUL movement over one, or only a few, CC-proximal cell types. Indeed, movement within leaves' lamina would primarily require amiRSUL translocation from the CCs to the phloem-parenchyma cells and bundle sheath (Aubry *et al*, 2019). Long-distance movement would involve, in principle, a single transloca-tion step given that CCs are directly connected to the SEs, upon which amiRSUL would likely be distributed to other organs follow-ing the phloem flow (Aubry *et al*, 2019). These CC-centric features of the *pSUC2::amiRSUL* system might have contributed to the strik-ing bias of the forward screen's outcome, with *hst* amounting to ~ 20% of isolated mutants with compromised amiRSUL silencing. Similarly, testing the HST dependency of endo-miRNA mobility involved, for technical feasibility, molecules that are all naturally expressed within, or in direct vicinity of, the vasculature. The leaf phenotype originally associated with *hst* also presumably involves miR390 movement from the vasculature to the surrounding tissues in young primordia, upon which specialized AGO7:miR390 complexes are proposed to trigger formation of a spatial gradient of mobile tasiRNAs underpinning leaf polarity (Schwab *et al*, 2009; Chitwood & Timmermans, 2010). This could explain, incidentally, why restricting HST's expression to the vasculature, using *pSUC2:: HST:GFP*, was sufficient to rescue the aberrant leaf shape of *hst-1* loss-of-function plants (Fig 5A; Appendix Fig S13A). Consistent with this idea, *AGO7* was identified in the same screen for juvenile-to-adult phase transition defects as *HST* (Hunter *et al*, 2003), and we have shown here that *hst* does not impede siRNA movement includ-ing, most likely, tasiRNA movement. We note, however, that the phyllotaxis defects of *hst-1* were not rescued by *pSUC2::HST:GFP*, suggesting that HST is required in additional, non-vascular tissues to carry out related or unrelated functions. Obtaining *in planta* information on HST's spatial expression would likely help address-ing some of the above issues. So far, however, all our attempts to generate functional transcriptional fusions to the presumptive *HST* promoter region have failed, which, incidentally, prompted the use of the *UBIQUITIN10* promoter in several parts of this study, as was also the case in the recent work of Cambiagno *et al* (2021).

### The expected mobile fate of AGO1-unbound as opposed to AGO1-bound miRNAs

Critical to an understanding of HST's role in regulating miRNA movement in addition to biogenesis, is the question of the molecular form(s) under which these molecules might move between cells and over long distances. As now established for siRNAs (Devers *et al*, 2020), our data support the currently prevailing view that processed miRNAs, not their precursors, are generally involved (Buhtz *et al*, 2010; Liu & Chen, 2018; Skopelitis *et al*, 2018; Brosnan *et al*, 2019). This could either implicate AGO-bound or AGO-unbound entities, whose existence has been recently demonstrated (Dalmadi *et al*, 2019). Of the two possibilities, we consider the former unlikely because the available data suggest that AGO proteins generally act cell autonomously (Chitwood & Timmermans, 2010; Melnyk *et al*, 2011; Skopelitis *et al*, 2018; Devers *et al*, 2020). This notion is supported by several lines of evidence in the specific case of AGO1, which is the main effector of mostly 5'U miRNAs including amiRSUL, endo-miR160, endo-miR395, and endo-miR165 studied

here. For instance, in five out of five root layers inspected in the *Arabidopsis* root tip, AGO1:GFP translational fusions expressed under individual root cell-specific promoters yielded florescent signals restricted within each cognate expression layers (Brosnan *et al*, 2019). This observed cell autonomy of AGO1 was, in fact, the very foundation for the development of the miRoot browser (https://www.miroot.ethz.ch/) enabling genome-scale comparative exploration of miRNA loading *versus* miRNA activity in space (Brosnan *et al*, 2019). Given the high volume load and unusually enlarged connections between CCs and SEs (Yan & Liu, 2020), we cannot formally rule out, however, that AGO1:miRNA complexes might leak into the phloem stream and, hence, contribute to long-distance transport. However, from the striking efficacies of amiRSUL and SO4⁻-induced miR395 graft transmissions (Figs 1E and 7A), this would likely involve substantial amounts of AGO1, yet the protein is systematically absent from comprehensive *Arabidopsis* phloem sap proteomes (Batailler *et al*, 2012; Guelette *et al*, 2012; Carella *et al*, 2016), as are indeed any other AGOs or any known RNA silencing components. Perhaps the most compelling evidence generally arguing against the movement of sRNAs bound to AGO1 is our recent finding that, as they move from silencing-emitting to silencing-recipient tissues, 5'U sRNAs are selectively retained within traversed cells (Devers *et al*, 2020), suggesting that their loading into cell-autonomous AGO1 antagonizes their movement. Indeed, the use of Meselect showed that compromising AGO1 loading, using, e.g., the strong *ago1-18* PAZ domain-mutant background, enhances 5'U sRNA movement (Devers *et al*, 2020). Analyzing the same samples revealed that *ago1-18* likewise promotes vasculature->epidermis movement of 5'U miR160 (Appendix Fig S16A), whereas using the same approach, we showed here that miR160 vasculature->epidermis movement is impeded in *hst-1* (Fig 7C and D). Therefore, HST's effect on movement unlikely involves AGO1-bound miRNAs, consistent with the loading efficacy of miRNAs into the global cellular pool of AGO1 remaining largely unchanged in the *hst* compared with WT background (Fig 3D).

**How might HST regulate miRNA movement?**

A role for HST in modulating general macromolecular trafficking is neither consistent with its subcellular localization (Figs 5C and D, and 6A) nor with the unaltered free GFP phloem-unloading pattern observed in *hst-1* compared with WT sink tissues (Fig 5B and E–G). HST's effects on miRNA movement likely entail a cell-autonomous activity of the protein because HST:GFP expressed from *pSUC2::HST:GFP* fully rescued amiRSUL movement while yielding a fluorescent signal strictly confined within the CCs (Fig 5C and D). The signal neither provided evidence for phloem unloading, unlike the signal from free GFP expressed from *pSUC2::GFP* (Fig 5E). Our results (Fig 6B and C; Appendix Fig S14) indicate that the cell-autonomous function of HST in controlling miRNA movement is unlikely to entail nucleo-cytosolic shuttling, consistent with the available literature also ruling out a role for HST in miRNA nuclear export (Bologna *et al*, 2018; Zhu *et al*, 2019; Zhang *et al*, 2020; Cambiagno *et al*, 2021). According to the arguments laid out in the previous section, HST's role in movement should mainly involve AGO1-unbound miRNAs. Such entities were recently discovered following observations that, at steady states, only a variable fraction of any given *Arabidopsis* miRNA is found within biologically active

AGO1-miRISCs, with the remaining non-loaded fraction not being attributed any specific role (Dalmadi *et al*, 2019). Both AGO1 protein availability and pri/pre-miRNA structural/sequence influence this loaded/unloaded-miRNA partitioning process (Dalmadi *et al*, 2019), which is thus expected to predominate within the nucleus. Indeed, the rate of NLS-dependent nuclear import and stability of AGO1 would influence the extent of miRNA loading, which is now admitted to prevail in the nucleus (Bologna *et al*, 2018; Zhang *et al*, 2020). Likewise, the abundance, processing rate, and processing accuracy of miRNA precursors transcribed in the nucleus would determine the quantity and quality of AGO1 cargoes involved in miRISC nuclear assembly before their export in a TREX-2- and XPO1-dependent manner (Bologna *et al*, 2018; Zhang *et al*, 2020). Within this overall context, the recent finding that HST modulates nuclear miRNA biogenesis by bridging *MIRNA* transcription with miRNA processing (Cambiagno *et al*, 2021) is of particular relevance to its requirement for miRNA movement, as discussed below.

In line with the effects of *crd1* in rice (Zhu *et al*, 2019), RNA sequencing analyses in *Arabidopsis* show that *hst-15* does not alter endo-miRNA processing *per se* (Cambiagno *et al*, 2021). We confirmed those results in *hst-1*, *hst-25*, and *hst-26* (Appendix Fig S16B), in contrast to the strong elevation in pri/pre-miRNA levels seen in the well-established miRNA processing-defective mutants *hyl1-2* and *se-1* (Appendix Fig S16B). Incidentally the former mutant had only a weak effect on amiRSUL movement from *pSUC2::amiRSUL^hyl1-2* scions to WT rootstocks, compared to *hst-1* (Fig 4C and D). Further consistent with a processing-independent role for HST in miRNA biogenesis, complete maturation of an artificial pri-miRNA occurred *in vitro* to the same extent with protein extracts from *hst-15* and WT plants, whereas it was strongly comprised with *dcl1* extracts, as expected (Cambiagno *et al*, 2021). A specific and robust interaction was nonetheless detected between HST and DCL1, but it was rationalized by further experiments showing that HST is likely required as a mere scaffold in this context, facilitating DCL1 recruitment onto genomic *MIRNA* loci via the MED37 complex involved in pri-miRNA transcription (Cambiagno *et al*, 2021).

The emerging positioning of HST in the nuclear miRNA biogenesis pathway is, in fact, ideally suited to support a model in which the protein would seize a variable fraction of DCL1-neo-processed miRNAs before their loading into AGO1. As discussed, nuclear loading of miRNAs would seal their cell-autonomous fate for the execution of intracellular silencing following their export in a TREX-2- and XPO1-dependent manner, as proposed (Bologna *et al*, 2018; Zhang *et al*, 2020). Consistent with this model, we detected amiRSUL, miR160, and miR165/166 in immunoprecipitates of HST:GFP ectopically overexpressed in *pUBQ::HST:GFP* plants (Appendix Fig S16C). However, only very small quantities of each miRNA were detected above background, suggesting either that non-loaded species constitute a minor fraction of the global nuclear pool of miRNAs or that HST interacts only transiently with non-loaded miRNAs. Of the two possibilities, we favor the latter, because AGO-unbound miRNAs are rather abundantly detected at least in total cell extracts (Dalmadi *et al*, 2019), and because robust (i.e., above background) detection of endo-miRNAs in HST immunoprecipitates requires prior cross-linking (Cambiagno *et al*, 2021), as expected from transient, as opposed to prolonged, HST:miRNA

interactions. Based on these findings, HST might thus transiently channel a fraction of non-loaded miRNAs into an as yet unidentified nuclear export pathway making them available in the cytosol. Once delivered into the cytosol, we anticipate that a possibly substantial fraction of these AGO-unbound miRNAs could eventually load into the neo-translated AGO1 to execute cell-autonomous silencing in a manner similar to nuclear-assembled and nuclear-exported miRISCs (Bologna *et al*, 2018; Zhang *et al*, 2020). However, a remaining fraction of cytosolic AGO-unbound miRNAs would be uniquely amenable to move into neighboring cells and over long distances after reaching the CC-SE interface.

The AGO-loaded *versus* non-loaded ratio may vary extensively from one miRNA species to another, moreover in a tissue-dependent manner (Dalmadi *et al*, 2019). Thus, according to the above model, some miRNAs are likely to engage HST more than others in the nucleus of only certain cell types. We hypothesize that, in the absence of HST specifying their nuclear exit, a given fraction of these non-loaded miRNAs might be degraded while another might ultimately load—or even overload—into nuclear AGO1 for silencing execution upon their TREX-2-/XPO1-dependent nuclear export. The likely variable nature of each fraction (depending on the miRNA under consideration) could potentially explain the tissue-dependent up/down fluctuations affecting the levels of some, but not other miRNAs in the *hst* background (Park *et al*, 2005; Cambiagno *et al*, 2021; this study). This scenario could also explain why the loading efficacy of amiRSUL into the global cellular pool of AGO1 was not overtly changed (Fig 3D). Indeed, the model predicts that, in the absence of HST, only the nuclear pool of AGO1 would possibly exhibit an increase in the loading of some, albeit not other, miRNAs, an issue requiring further investigation. Contributing further to these proposed subtle and complex effects of the *hst* mutation is the observation that only some miRNAs appear to move, including to substantially varying extents depending on the miRNA species under consideration (Brosnan *et al*, 2019).

A nuclear role for HST in specifying miRNA movement at a step linked to *MIRNA* transcription (by MED37) and miRNA processing (by DCL1)—as recently proposed by Cambiagno *et al* (2021) to implicate HST in miRNA biogenesis—is further consistent with observations made with siRNAs. Indeed, neither mobile S-PTGS triggered by the L1 locus (Park *et al*, 2005) nor the movement/activity of siRNAs derived from the *SUC-SUL* or other *IR* loci (Fig 6D) (Park *et al*, 2005) was impeded in *hst*. Unlike miRNAs, however, endogenous and transgenic siRNAs are synthesized and/or loaded in the cytosol, moreover via a completely distinct machinery (Jouannet *et al*, 2012; Ye *et al*, 2012; Pumplin *et al*, 2016; Bologna *et al*, 2018). This would render their movement *de facto* HST independent. Secondly, silencing execution in miRNA-recipient cells would require the delivered AGO-unbound miRNAs to load into AGO1. As suggested for cytosolic tasiRNAs, this would likely involve the recipient cells' cytosolic pool of neo-translated AGO1, before its NLS-dependent nuclear import (Bologna *et al*, 2018). Alternatively, and although no experimental evidence supports this theoretical possibility, the miRNAs delivered in recipient cells could be potentially imported into the nucleus to be subsequently loaded into the nuclear pool of AGO1. The neo-constituted miRISCs would then be exported to the cytosol in a TREX-2- and XPO1-dependent manner (Bologna *et al*, 2018; Zhang *et al*, 2020). In neither of these two situations, however, mobile miRNAs would intercept HST because its

action is linked to *MIRNA* transcription/processing (Cambiagno *et al*, 2021), which would only occur in silencing-emitting cells. This conjecture is consistent with our findings, here, that (i) selectively and cell autonomously expressing HST:GFP in amiRSUL-emitting CCs suffices to fully restore amiRSUL movement in leaves with the *hst-1* mutant background (Fig 5A) and that (ii) HST is dispensable for the normal execution of amiRSUL-mediated silencing in grafted amiRSUL-recipient rootstocks (Fig 3F).

The present study identifies a *hitherto* unrecognized role for HST in miRNA movement, which, we contend, is coordinated with its unique and singular contribution to miRNA biogenesis in the nucleus. At this stage, however, it would be premature to exclude an additional role for HST in the cytosol of miRNA-emitting cells. This possibility, left open by our analysis of hst-3:GFP (Fig 6C, Appendix Fig S14), can now be experimentally tested given the recent availability of the mainly cytosolic HST$^{\Delta N}$ allele and, conversely, the HST$^{ran1}$ background in which HST is mainly nuclear (Cambiagno *et al*, 2021). It is also anticipated that any other factor modulating the loading of miRNAs into the cytoplasmic pool of AGO1, after their nuclear export, will likely impact their movement. A final pressing and fascinating issue, already evoked elsewhere (Devers *et al*, 2020), pertains to the molecular feature(s) that should disqualify a neo-processed miRNAs from loading into AGO1 and concurrently license the molecule for movement in a HST-dependent manner. In principle, such feature(s) should not only manifest in the sRNA-emitting cells, but it should also be reversed in the sRNA-receiving cells in which silencing execution requires AGO1 loading.

# Materials and Methods

### Plant material and growth conditions

All *Arabidopsis thaliana* plants used in this study were in the Col-0 ecotype background. The *hst-1*, *hst-3*, *ago1-18*, *ago1-27*, *hyl1-2*, *se-1*, *dcl2-1*, *dcl3-1*, *dcl4-2*, *hen1-6*, and *shr-2* mutant lines and *pSUC2::GFP*, *pSUC2::tmGFP9*, and *SUC-SUL* transgenic lines were described previously (Fukaki *et al*, 1998; Telfer & Poethig, 1998; Imlau *et al*, 1999; Prigge & Wagner, 2001; Morel *et al*, 2002; Bollman *et al*, 2003; Himber *et al*, 2003; Vazquez *et al*, 2004a; Xie *et al*, 2004; Li *et al*, 2005; Sorin *et al*, 2005; Stadler *et al*, 2005; Xie *et al*, 2005). Surface sterilized seeds were sown on ½MS medium containing MES buffer and vitamins (Duchefa Biochemie) and solidified with 0.8% microagar. Plants were cultivated *in vitro* at 21°C in 12-h light/12-h dark conditions for two weeks before transplanting in soil. Further growing was done at 21°C in 16-h light/8-h dark conditions. Light intensity was 120 $\mu$E.m$^{-2}$.s$^{-1}$ in every condition. Plant phenotype pictures were taken on 4-week-old plants, with the exception of *in vitro* cultured plants, for which phenotyping pictures of 2-week-old plants were taken using a M205 FCA fluorescence stereo microscope (Leica).

### Cloning procedures / genotyping

Unless specified, DNA cloning was done using Phusion High-Fidelity DNA Polymerase for PCR amplifications and the Gateway cloning technology (Thermo Fisher Scientific). *SUC2* (At1g22710) and *UBQ10* (At4g05320) promoter sequences, as well as *HST* (At3g05040) coding

sequence, were PCR-amplified from *Arabidopsis thaliana* genomic DNA (promoters) or leaf cDNA (*HST*) using primers listed in Appendix Table S1. After agarose gel purification, attB-flanked DNA fragments were inserted by BP recombination into the pDONR P4-P1r donor vector for promoters or into the pDONR221 for *HST*, giving rise to the entry vectors pENTR_attL4-*pSUC2*-attR1, pENTR_attL4-*pUBQ*-attR1, and pENTR_attL1-*HST*-attL2. pENTR_attL1-*hst-3*-attL2 mutant entry vector was obtained by PCR-based site-directed mutagenesis of pENTR_attL1-*HST*-attL2 using primers listed in Appendix Table S1. The *pSUC2::amiRSUL* construct was obtained by recombining the pENTR_attL4-*pSUC2*-attR1 vector with an attL1-attL2 entry vector that contains the *MIR319a* backbone modified to produce the miRNA UUAAGUGUCACGGAAAUCCCU targeting the *SUL* homolog *CH42* (At4g18480) (de Felippes *et al*, 2011) into the binary vector pB7m24GW (Karimi *et al*, 2007). The *pSUC2::GUS* construct was obtained by recombining pENTR_attL4-*pSUC2*-attR1 with an attL1-attL2 entry vector containing a 2xFLAG-2xHA epitope tag coding sequence together with an attR2-attL3 entry vector containing the β-glucuronidase (GUS) coding sequence into the binary vector pB7m34GW (Karimi *et al*, 2007). The *pSUC2::HST:GFP*, *pUBQ::HST:GFP*, and *pUBQ::hst-3:GFP* constructs were obtained by shuttling the *pSUC2* or *pUBQ10* promoter sequences with the *HST* or *hst-3* coding sequence, together with the *eGFP* coding sequence cloned in an attR2-attL3 entry vector, into the binary vector pK7m34GW (Karimi *et al*, 2007). The resulting binary vectors were introduced into the *Agrobacterium* strain GV3101 to transform WT or mutant *Arabidopsis* plants by the floral dip method (Clough & Bent, 1998). T1 primary transformants were selected either in soil using BASTA (*pSUC2::amiRSUL* and *pSUC2::GUS*) or on ½MS plates containing kanamycin (*pSUC2::HST:GFP*, *pUBQ::HST:GFP*, and *pUBQ::hst-3:GFP*). T2 plants were assessed for single-locus insertions (3:1 segregation ratio) before propagation to homozygous T3 generations. In addition, using Southern blot analysis and TAIL-PCR (Liu *et al*, 1995), the precise genomic insertion of the *pSUC2::amiRSUL* transgene was located in the intergenic region between At3g19660 and At3g19663. Primers for genotyping the *pSUC2::amiRSUL* transgene are listed in Appendix Table S1.

### EMS mutagenesis and mutation mapping

The *pSUC2::amiRSUL* reporter line was mutagenized according to Weigel and Glazebrook (2002) with minor modifications. Seeds were washed in 0.1% Tween during 15 min before incubation for 12 h in 0.25% ethyl methane sulfonate (EMS) (Merck Sigma) at 21°C. Seeds were extensively washed with distilled water for 4 h before sowing on soil. Resulting M1 plants were left for self-fertilization, and M2 seeds were collected from individual M1 plants. Mutants in which the *pSUC2::amiRSUL* vein-chlorosis phenotype was reduced were screened among 1,750 M2 progenies. Potential candidates were back-crossed once to the parental reporter *pSUC2::amiRSUL*. One hundred F2 segregating mutant plants (2 weeks old) were then harvested and pooled for genomic DNA extraction following a CTAB DNA extraction (Clarke, 2009), in parallel with the *pSUC2::amiRSUL* parental line. Whole-genome resequencing of parental and pooled mutant DNA was performed using a TruSeq Nano DNA Library Prep kit and a HiSeq4000 sequencing system (Illumina) to get an average genome coverage of 50X. Mutant DNA SNPs were mapped onto the *Arabidopsis* genome and filtered against the parental DNA SNPs using CLC Genomics

Workbench (Qiagen) resequencing tools. Putative causal mutations were restricted to EMS transition mutations found in 100% of the sequencing reads and inducing amino acid changes or splice site effects in coding sequences.

### *Arabidopsis* micrografting procedure and sulfate starvation

Micrografting was done essentially as described earlier (Andersen *et al*, 2013). Briefly, 5- to 7-day-old seedlings grown vertically on ½MS plates were transferred onto a MF-Millipore membrane filter (3 µm pore size, Merck) placed on two layers of Whatman 3MM paper wet with sterile water in a petri dish. Scions were processed by removing both their cotyledons and cutting their hypocotyl within a millimeter to the shoot apex with a surgery blade No. 15. Rootstocks were similarly cut in their hypocotyls, then aligned to the scions. Plates were subsequently sealed with parafilm and kept vertically for 7 days, under *in vitro* culture conditions. Grafted plants were transferred on ½MS medium for 2 weeks growth, after removal of plants with visible adventitious roots. In experiments involving sulfate starvation of plants, plantlets were further grown for 96 h on a modified ½MS medium, in which sulfate salts were replaced by their equivalent chloride salts, solidified with 0.8% molecular biology grade agarose.

### Separation of vascular and epidermal tissues from *Arabidopsis* leaves using Meselect

Meselect was carried out mainly as described in Svozil *et al* (2016) with a few modifications. After placing the leaf in a tape sandwich, the lower epidermis was peeled away from the vasculature and the other upper leaf tissues. The epidermis tape was directly frozen in liquid nitrogen while the vasculature tape was incubated in protoplasting solution composed of 1% cellulase Onozuka R-10 (Serva), 0.25% macerozyme R-10 (Serva), 0.4 M mannitol, 10 mM CaCl$_2$, 20 mM KCl, 0.1% (wt/vol) BSA, 20 mM MES pH 5.7 for about 30 min at room temperature with gentle agitation, until the vasculature of the leaf petiole started to detach from the tape. Leaf vasculature tissue was pulled from the tape with forceps, washed twice in washing buffer (154 mM NaCl, 125 mM CaCl$_2$, 5 mM KCl, 2 mM MES, pH 5.7), and frozen in liquid nitrogen. Epidermal and vasculature tissues were ground into a fine powder with liquid nitrogen before proceeding to RNA extraction.

### Infestation with aphids

Green peach aphids (*Myzus persicae*) were maintained on turnips (*Brassica rapa*, Tokyo cultivar) in a dedicated chamber at 23°C with 12-h light/12-h dark conditions. Around 20 adult aphids were applied onto the rosettes of bolting *Arabidopsis* plants in a confined chamber at 21°C with 16-h light/8-h dark conditions. The insect population was collected 14 days after infestation and immediately ground in fine powder with liquid nitrogen before further analysis.

### Immunoprecipitation (IP) experiments

In AGO1-IP and HST-GFP-IP experiments, 4-week-old *Arabidopsis* rosettes ground in liquid nitrogen were resuspended in 3 ml for 1 g of tissues powder in IP buffer (50 mM Tris–HCl pH 7.5, 150 mM

NaCl, 10% glycerol, 0.1% NP-40), containing 2 µM MG-132 and one tablet of cOmplete® protease inhibitor cocktail (Merck Roche) per 10 ml. All further steps were carried out on ice or in a 4°C cold chamber. After 10 min of gentle mixing, lysates were cleared from cell debris twice by centrifugation at 16 000 $g$ for 15 min. 45 µl of cleared supernatants was mixed to 4× Western blot loading buffer for further analysis of input protein fractions. In addition, 200 µl was collected for RNA extraction. 1 ml of cleared lysates was subsequently used for AGO1- or HST-GFP experiments. For AGO1 IPs, lysates were first pre-cleared with 40 µl of protein A agarose beads (Merck Roche) for 1 h on a rotating wheel. Pre-cleared lysates were then incubated with 1 µl of anti-AGO1 antibody (Agrisera, ref. AS09 527) for 1 h under gentle mixing, followed by the addition of 40 µl of protein A agarose beads and another incubation for 1 h with gentle agitation. For HST-GFP IPs, lysates were incubated for 1 h on a rotating wheel with 30 µl of GFP-trap magnetic agarose beads (Chromotek), pre-blocked with 2% BSA in IP buffer. Agarose or magnetic bead conjugates were washed 3 times with IP buffer for 10 min, collected, and resuspended in 500 µl of TRI Reagent (Merck) for RNA and protein extraction, according to the manufacturer's instructions. Immunoprecipitated RNA was precipitated from the aqueous phase with addition of 20 µg of glycogen. Immunoprecipitated proteins were retrieved from 200 µl of the organic phase by addition of 1 ml of 0.1 M ammonium acetate in methanol, followed by 1 h incubation at −20°C. Proteins were pelleted by centrifugation at 16 000 $g$ for 20 min, washed twice with 0.1 M ammonium acetate in methanol, and resuspended in 1× Western blot loading buffer (10% glycerol, 4% SDS, 62.5 mM Tris–HCl pH 6,8, 5% 2-mercaptoethanol). RNA from input samples was extracted by adding 1 volume of Roti®Phenol/Chloroform/Isoamyl alcohol (Carl Roth), precipitated from the aqueous phase with 1 volume of isopropanol in the presence of 0.3 M sodium acetate pH 5.2, and washed with 80% ethanol before resuspension in Northern blot loading buffer.

### RNA extraction and northern analysis

RNA was extracted from frozen tissues ground in liquid nitrogen using TRI Reagent (Merck) according to the manufacturer's instructions and resuspended in water. Equal amounts of RNA (1 to 10 µg), dried with a vacuum concentrator, or immunoprecipitated RNA fractions were resuspended in Northern blot loading buffer (50% formamide, 10% glycerol, 10 mM Tris pH7.7, 1 mM EDTA, 0.01% bromophenol Blue), resolved by electrophoresis on a denaturating polyacrylamide gel (0.5X TBE, 17.5% acrylamide/bisacrylamide 19:1, 8 M urea), transferred on a Hybond-NX Nylon membrane (Merck Sigma) in 0.5X TBE, and cross-linked using 1-ethyl-3-(3-dimethylaminopropyl)carbodiimide (EDC) according to Pall and Hamilton (2008) for 2 h at 60°C. Membranes were incubated in PerfectHyb Plus Hybridization buffer (Merck Sigma) at 42°C overnight with an oligonucleotide probe complementary to a specific miRNA sequence and 5'-end-labeled with [γ-$^{32}$P]ATP using T4 PNK (Thermo Fisher Scientific). Oligonucleotide probe sequences are listed in Appendix Table S1. Membranes were washed 3 times for 15 min with 2X SSC, 2% SDS at 42°C and exposed overnight to a storage phosphor screen, which was subsequently imaged on a Typhoon FLA9000 (GE Healthcare). Band quantifications were done using Image Lab software (Bio-Rad) with auto-contrasted images (0.00% clipping values for both shadows and highlights). For sequential hybridizations of probes, membranes were stripped in 0.1% SDS at 90°C three times for 15 min before re-probing with labeled oligonucleotides as described above.

### RT–PCR, RT–qPCR, and stem-loop RT–qPCR

For RT–PCR and RT–qPCR, 2 µg of RNA extracted with TRI Reagent was treated with 1 unit of DNAse I (Thermo Fisher Scientific) for 30 min at 37°C and reverse-transcribed with the RevertAid First Strand cDNA Synthesis Kit (Thermo Fisher Scientific) using a poly-dT primer, according to the manufacturer's instructions. Stem-loop RT–qPCR was carried out essentially according to Varkonyi-Gasic *et al* (2007), using the RevertAid First Strand cDNA Synthesis kit and by multiplexing stem-loop RT primers listed in Appendix Table S1. In RT–PCR experiments, DreamTaq polymerase (Thermo Fisher Scientific) was used together with 1 µl of cDNA in 20 µl PCRs, according to the manufacturer's instructions. *amiRSUL* primary transcript (*pri-amiRSUL*), *HST*, and control *TCTP* (At3g16640) (Brioudes *et al*, 2010) cDNAs were amplified using primers listed in Appendix Table S1 with 40, 28, and 22 PCR cycles, respectively. PCR products were resolved by electrophoresis on a 1% agarose gel containing ethidium bromide for imaging. In RT–qPCR and stem-loop RT–qPCR experiments, 1 µl of cDNA was used in 10 µl PCRs containing KAPA SYBR FAST qPCR 2X master mix (Merck Sigma) and gene-specific or miRNA-specific primers (0.2 µM each) listed in Appendix Table S1. qPCRs were performed in triplicates in 384-well plates using a LightCycler 480 System (Roche) and following the PCR program recommended with the KAPA SYBR FAST qPCR mix. In addition, a melting curve was performed to verify the specificity of each PCR amplification. Cp values (cycle values of the maximum second derivative of the amplification curves) were calculated for each PCR with the LightCycler 480 software. Relative expression values for each mRNA, pri-miRNA, or miRNA were obtained by calculating $2^{-\Delta Cp}$, where ΔCp represents the difference between the Cp value of the analyzed RNA and the mean of the Cp values of (i) *ACT2* (At3g18780), *RHIP1* (At4g26410), and *YLS8* (At5g08290) control mRNAs in RT–qPCR experiments, or (ii) *snoR85* (At1g09873) and *U6* (At3g14735) small nucleolar RNAs in stem-loop RT–qPCR experiments. Relative expression values from independent distinct samples were normalized with the mean of the control condition values and further individually plotted on graphs, together with their mean and standard deviation (SD). Normality distribution and homoscedasticity of the expression values were tested with Shapiro–Wilk and Fisher tests, respectively. Unpaired two-sided *t*-tests (with Welch's correction in case of heteroscedasticity), as well as 1-way or 2-way ANOVA followed by Tukey's multiple comparisons tests, were applied to compare the means of the expression values. All statistical analyses were carried out using GraphPad Prism software.

### Protein extraction and western blotting

Except for immunoprecipitation experiments, total plant proteins were isolated from 4-week-old leaves by phenol extraction as described in Schuster and Davies (1983), with some modifications. Plant tissues ground in liquid nitrogen were resuspended in 0.7 M sucrose, 0.5 M Tris–HCl pH8, 5 mM EDTA, 0.1 M NaCl, 2% 2-mercaptoethanol, and cOmplete® protease inhibitor cocktail (Merck

Roche, one tablet per 10 ml). One volume of Roti®Phenol (Carl Roth) was added and the mixture shaken for 10 min at room temperature. The phenol phase was recovered after centrifugation at 16 000 $g$ for 10 min at 4°C, and proteins were precipitated by addition of 5 volumes of 0.1 M ammonium acetate in methanol, followed by 1 h of incubation at −20°C. Proteins were pelleted by centrifugation at 16 000 $g$ for 20 min at 4°C and washed twice with 0.1 M ammonium acetate in methanol before resuspension in 3% SDS, 62.3 mM Tris–HCl pH 8, 10% glycerol. Total protein concentrations were estimated using NanoDrop A280 measurement and normalized quantities of proteins mixed to 4× Western blot loading buffer. For GFP, AGO1, or HST western analysis, proteins were separated by SDS–PAGE and transferred onto immobilon-P PVDF membranes (Merck Millipore). Membranes were blocked for 30 min in 1× PBS supplemented with 1% BSA and subsequently incubated with primary antibodies overnight at 4°C in the same solution. The monoclonal anti-GFP (Chromotek, ref [3H9]) and polyclonal anti-AGO1 antibodies (Agrisera, ref. AS09 527) were diluted at 1:8,000. The polyclonal anti-HST antibody was raised in rabbit using the peptide H$_2$N–EFEGKGDFGPYRSKLC–CONH$_2$ as antigen (Eurogentec) and diluted at 1:2,000 for western analysis. Membranes were washed 3 times with PBS-T (1× PBS + 0.1% Tween-20), incubated for 1 h at room temperature with 1:10 000 dilutions of HRP-conjugated goat anti-rat (GFP western analysis) secondary antibody (Abcam, ref. ab6845) or HRP-conjugated goat anti-rabbit (AGO1 and HST western analysis) secondary antibody (Thermo Fisher Scientific, ref. 65-6120) in PBS-T 1% BSA, and washed again 3 times with PBS-T. Protein detection was carried out with the Westar Supernova ECL substrate (Cyanagen) and imaged via the ChemiDoc Touch Imaging System (Bio-Rad). Membranes were stained with Coomassie blue to reveal total proteins. For SUL and ACT western analysis, proteins were separated by SDS–PAGE and transferred to immobilon-FL PVDF membranes (Merck Millipore). Membranes were blocked for 30 min in 1× TBS supplemented with 5% skim milk powder, then incubated overnight at 4°C with a 1:8,000 diluted polyclonal anti-SUL antibody produced in rabbit (Brodersen et al, 2008), together with a 1:16,000 diluted monoclonal anti-actin (plant) antibody produced in mouse (Merck Sigma, Ref. A0480) in TBS-T (1× TBS + 0.1% Tween-20) 5% milk. Membranes were washed 3 times with TBS-T followed by incubation for 1 h at room temperature with IRDye 800CW Donkey anti-Rabbit and IRDye 680RD Donkey anti-Mouse secondary antibodies (Li-Cor) both 1:20,000 diluted in TBS-T 5% milk 0.01% SDS. Membranes were washed again 3 times with TBS-T and twice with TBS before drying. Protein detection was carried out by using an Odyssey CLx imaging system (Li-Cor) with automatic scanning settings. Protein band quantification was done with Image Lab software (Bio-Rad) using auto-contrasted images. Membranes were stained with Coomassie blue to reveal total proteins.

### Plant tissue staining procedures and confocal microscopy

GFP-tagged protein localization in companion cells of *Arabidopsis* first leaves and root differentiation zones was analyzed in 14-day-old plants grown *in vitro* on 1/2 MS plates. Seedlings were cleared using the ClearSee method (Kurihara et al, 2015). Briefly, plantlets were fixed with 4% PFA in 1× PBS for 1 h at room temperature under vacuum, washed twice in 1× PBS, cleared for 1 week in the ClearSee solution (10% xylitol, 15% sodium deoxycholate, 25% urea), with regular clearing solution changes, stained with 0.01% Calcofluor White (CW) overnight, and washed with the ClearSee solution for 1 h before confocal imaging. GFP-tagged protein localization in differentiation and division zones of *Arabidopsis* roots was studied in 7-day-old plants grown vertically *in vitro* on ½MS plates. Living roots were stained for 10 min with 10 μg/ml propidium iodide (PI) in water before confocal imaging. Basic Fuchsin staining of *Arabidopsis* roots was performed with 7-day-old plants grown *in vitro* on ½MS vertical plates. Roots were cleared in 1 M KOH for 6 h at 37°C, then stained with 0.01% basic fuchsin for 10 min under gentle agitation. Roots were destained overnight in 70% ethanol and rehydrated in water before confocal imaging.

Confocal pictures were acquired using a Zeiss LSM 780 microscope controlled by the Zeiss Zen software. 488-nm excitation laser was used for GFP imaging, together with 500–550 nm emission detection band. CW-, PI-, and basic fuchsin-stained tissues were imaged using 405, 488, and 561 nm excitation lasers, respectively, together with 425–475 nm, 620–720 nm, and 600–700 nm emission detection bands, respectively. Confocal image adjustments and, when required, z-stack projections were further carried out using NIH ImageJ software.

### Bioinformatic analyses

Appendix Fig S10A was generated based on processed data from Cambiagno et al (2021). Raw read count values of Col-0 (WT) and *hst-15* replicates in the Supplemental Table 2 of Cambiagno et al (2021) were averaged, transformed into pseudo-count by adding +1, and used as input for scatterplot representation. miRNAs with significance values (FDR.DE) lower than 0.05 were considered as differentially accumulated.

Trimming, tailing, and isoform estimates (Appendix Fig S10B) were calculated with a method inspired from Giudicatti et al (2021), which identifies potentially unaltered, trimmed, or tailed miRNAs, and assigns an index by dividing the number of trimmed or tailed molecules with the one of the annotated (unaltered) miRNA sequences. Raw sequencing data of *hst-15*, *hen1*, and *hen1 heso1* were obtained from SRA (accessions ERP126434 and SRX3405447). Only R1 reads in the paired-end sequencing data of ERP126434 were used. When needed, adapter sequences were removed using fastx_clipper from http://hannonlab.cshl.edu/fastx_toolkit/ with options "-a TGGAATTCTCGG-l 15-c-Q 33". Trimmed reads with similar sequences were grouped using the processReads function from the ncPRO-seq pipeline (Chen et al, 2012) and mapped onto *Arabidopsis* TAIR10 reference genome using bowtie (Langmead et al, 2009) (v1.2.3; options -v 0 -a -m 500 --best --strata --nomaqround -y --phred33-quals) for isoform estimate or Bowtie 2 (Langmead & Salzberg, 2012) (v2.4.1; options -k 100) for trimming and tailing estimates, considering that Bowtie 2 allows higher mismatches/gaps, expected for modified miRNA sequences. Tailing and trimming estimates were carried out by retrieving 18 to 26-nt-long reads with a 5' position starting exactly at the 5' position of annotated miRNAs. Reads corresponding exactly to the mature miRNA sequences as well as shorter or longer reads were counted for each miRNA. The log ratios shorter reads/mature miRNA and longer reads/mature miRNA were then calculated and represented as boxplot using jitter in R.

Alternative processing estimate (isoform estimate) was performed for each miRNA by retrieving reads whose alignment is nested in the mature miRNA annotation enlarged by +/− 5 nucleotides. Perfectly aligned reads (no mismatch nor gaps) with length corresponding to the one of the mature miRNA +/− 1-nt were counted. The number of reads corresponding exactly to the annotated mature miRNA sequence was subtracted in order to determine the number of other remaining reads, which could correspond to misprocessing or alternative processing events. For each annotated miRNA with at least 5 reads in all the replicates of at least one genotype, the log ratios mature miRNA/other reads were calculated and represented as boxplot with jitter in R. Appendix Fig S10C and D were obtained by crossing the differential analysis results of WT and *hst-15* RNA sequencing data available in the Supplemental Table 3 of Cambiagno *et al* (2021) with the list of all *Arabidopsis* miRNA targets as compiled in Arribas-Hernandez *et al* (2016).

## Data availability

This study includes no data deposited in external repositories.

**Expanded View** for this article is available online.

## Acknowledgements

We thank members of the Voinnet laboratory for scientific input and critical reading of the manuscript. We particularly thank C.A. Brosnan for his advices in grafting and peeling experiments and A. Imboden for assistance in plant growth. We thank F.F. de Felippes and P.E. Jullien for providing plasmid vectors, C. Vorburger for providing green peach aphids, and P.E. Jensen for providing anti-SUL antibody. We acknowledge support of the Scientific Center for Optical and Electron Microscopy (ScopeM) of the ETH-Z, as well as support of the Functional Genomics Center Zurich (FGCZ) of the ETH-Z and the University of Zurich. The research work was supported by an ETH-Z post-doctoral grant attributed to F.B. and a European Research Council Advanced Grant (Frontiers in RNAi-II No. 323071) attributed to O.V.

## Author contributions

FB and OV designed the study; FB and FJ conducted all experiments; AS, TG, and ED contributed to biological resources and methodology; FB and OV analyzed the data and wrote the manuscript.

## Conflict of interest

The authors declare that they have no conflict of interest.

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
