## [Review Process File · The EMBO Journal]

HASTY, the Arabidopsis EXPORTIN5 ortholog, regulates cell-to-cell and vascular microRNA movement

Florian BRIOUDES, Florence JAY, Alexis Sarazin, Thomas Grentzinger, Emanuel Devers, and Olivier Voinnet

DOI: [10.15252/embj.2020107455](https://doi.org/10.15252/embj.2020107455)

Corresponding author: Olivier Voinnet (voinneto@ethz.ch)

Review Timeline:

Submission Date:	18th Dec 20
Editorial Decision:	9th Feb 21
Revision Received:	30th Mar 21
Editorial Decision:	28th Apr 21
Revision Received:	4th May 21
Accepted:	7th May 21

Editor: Ieva Gailite

Transaction Report:

Thank you for submitting your manuscript for consideration by the EMBO Journal. I sincerely apologise for the protracted review process due to delayed submission of reviewer reports and the currently high manuscript submission rate to our office. We have now received two referee reports on your manuscript, which are included below for your information.

As you will see from the comments, while reviewer #1 is more positive in their assessment, reviewer #2 indicates that clearer analysis of the relative contribution of HST to miRNA production vs miRNA nucleocytoplasmic shuttling would be required, and also points out that the published studies currently support the role of HST in miRNA biogenesis. Since the molecular details of how HST contributes to systemic miRNA transport remain vague at this point, I agree with reviewer #2 that this issue has to be clarified in the revised version in order to differentiate between a passive effect of miRNA level changes in the emitting cells vs an active contribution of HST to miRNA systemic transport. I would therefore like to invite you to address the comments of both referees in a revised version of your manuscript, while particularly focusing on this aspect. Please note that a strong referee support will be required for the acceptance of the revised manuscript.

I should add that it is The EMBO Journal policy to allow only a single major round of revision and that it is therefore important to resolve the main concerns at this stage. I would be happy to discuss the revision in more detail via email or phone/videoconferencing.

We have extended our 'scooping protection policy' beyond the usual 3 month revision timeline to cover the period required for a full revision to address the essential experimental issues. This means that competing manuscripts published during revision period will not negatively impact on our assessment of the conceptual advance presented by your study. Please contact me if you see a paper with related content published elsewhere to discuss the appropriate course of action.

Please feel free to contact me if you have any further questions regarding the revision. Thank you for the opportunity to consider your work for publication. I look forward to receiving your revised manuscript.

Referee #1:

This is an excellent paper that finally explain the role of HST in the miRNA pathway and reconcile all existing data. The authors show that HST (the plant XPO5 ortholog) is required to export miRNAs from emitting tissues, but not for miRNA reception or miRNA activity in recipient tissues. Their results are very clear and convincing, and what they found using a system based on an artificial miRNA applies also to endogenous miRNAs. The paper is well written and discussed, and represents a real advance in our understanding of the plant miRNA pathway.

I only have a few suggestions:

The authors show the reduction of SUL mRNA accumulation in leaves of pSUC2::amiRSUL plants and in roots of wt plants grafted to pSUC2::amiRSUL scions, but not in roots of pSUC2::amiRSUL plants, which should be the control for Fig1D. Is the reduction of SUL mRNA accumulation as

efficient with amiRSUL transmitted to the roots as it is with amiRSUL produced in the roots ?

The effect of *se-1* is not convincing, and should be removed. The effect of *ago1-27* is limited, thus the authors cannot say that silencing is suppressed. Maybe they could remind the readers what is the effect of the *ago1-27* mutation.

The authors correctly report that *hst* does not suppress transgene S-PTGS (Park et al, 2005), indicating that HST is not required for the movement of siRNA, but they missed that *hst* in fact enhances transgene S-PTGS (Martinez de Alba et al, 2011). Given the competition between miRNA and siRNA for AGO1, maybe the authors could take this into account in their discussion.

Other minor comments:

One sentence in the introduction appears truncated: "Whether exosomes or other vesicles are used in sRNA trafficking between plant cells remains unknown, however, given that plant cells are separated by cell walls."

The authors should not use nomenclature like pSUC2::amiRSULhen1 in the text and pSUC2::amiRSUL x hen1 in their figures (it looks like they are analyzing F1 plants). Instead they should use regular nomenclature pSUC2::amiRSUL hen1 in both text and figures

Referee #2:

This manuscript describes HASTY's identification and characterization as a critical player during non-cell-autonomous miRNA movement, both at short and long-distance, in plants. Using a genetic approach, the authors found several alleles of *hst* in a screening designed to identify mutants deficient in miRNA movement. Later on, the authors showed that the molecules moving are likely mature miRNA rather than the precursors. They also found that it is unlikely that AGO1 bound miRNAs are the moving entity pointing to the free cytoplasmic portion of miRNAs as the best candidate for non-cell-autonomous effect. Similarly, they found that HST itself is not moving out of the cell and thus, whatever effect it has on miRNA movement is cell-autonomous. Methodologically the paper is solid, and the experiments are well designed and performed. Even when I only have a few concerns about the experimental part itself, I believe the papers' proposed model and conclusions are biased, not supported by the evidence provided and unnecessarily opposed to the current knowledge of HST function. For example, one of the central premises in their construction of the story is that miRNA production is not affected in *hst* mutants. Probably the authors took this path, as a reduction in miRNA production will largely explain most of the results and observations. The provided evidence does not refute the impact of *hst* mutation on miRNA production. The authors need to better fit their model and results to existing evidence that suggests that HST significantly impacts miRNA accumulation. Some evidence (especially figure 6) is strong enough to demonstrate an effect of HST on miRNA movement, even if it also has a role in miRNA accumulation.

Similarly, and in the absence of a molecular mechanism that explains how HST control miRNA movement, the authors rely on the assumption that HST has a significant role in miRNA shuttling from nucleus to cytoplasm that impacts miRNA movement. Evidence provided by numerous authors, including themselves in previous papers and this one, has refuted a potential role of HST on miRNA shuttling (see below). Authors need to either provide convincing evidence of the premises mentioned above or present a less speculative model based on available data. A recent

paper offers new evidence on HST functions and needs to be considered here; probably the authors were not aware of this publication by the submission time as it is quite new.

-L42-45: it would be a good idea to introduce in this sentence also CRD1 (10.1111/tpj.14445), HASTY ortholog in rice as its molecular features and effects upon mutation are shared with Arabidopsis.

-L47, a non-affected miRNA Nuclear/Cytoplasmic partition in different hst alleles was also observed at least by (10.1038/s41477-020-0726-z) and (10.1016/j.molp.2020.12.019) in Arabidopsis and by (10.1111/tpj.14445) in rice. It was also observed in a previous report of the authors (10.1016/j.molcel.2018.01.007, Figure 4C) where they found no over-accumulation of nuclear miRNAs in hst alleles by FACS. Even when you can argue that in the absence of HST miRNA can be loaded in AGO1 and exported by this mechanism compensating the lack of HST, this idea is refuted by this article when you showed AGO1-miRNA loading does not increase in hst mutants (figure 3B).

-L169-178: Can you dismiss this possibility by measuring pri-miRSUL content in the phloem, as done in figure 1F for mature amiRSUL, using aphids?

-Please change EXPORTIN5 by HASTY, which is the correct name in Arabidopsis, in the title. If you want to mention EXP5 to highlight its orthology with the metazoan gene you can go for: HASTY, the Arabidopsis ortholog of EXPORTIN5, regulates the movement of microRNAs

-L125-127: perhaps it is better to use the word impaired or reduced instead of suppressed, as bleaching is still observable in several of these mutants.

-L198: I'm curious why hst-25, which is very similar regarding the mutation to hst-1, does not share a similar phenotype even when the protein is undetectable in both alleles (Figure 2B). Have you cloned and sequenced the two lower bands observed in figure S5 for hst-25 to address if some of these are the correctly spliced mRNA or a version retaining the ORF?

-The authors argue that hst mutants have no impaired miRNA production in part based on the fact that hst-25 and -26 still have some degree of vein chlorosis. Although this is true, the reduction is notorious and compatible with the previously reported decrease in miRNA production. As a matter of fact, the lack of chlorosis observed in these plants (figure 2C) is even stronger than in canonical biogenesis mutants such as hyl1 and se, or miRNA turnover mutants such as hen1 (Figure S1C). The complete lost of chlorosis in hst-1 (fig 2C) also point in the direction of a robust cell-autonomous effect in the mutant on miRNA production.

-On the paragraph starting in line 214: It is impossible to address whether the amiRSUL in figure 2F is enough to saturate the system and remain active. Conversely, is not clear here whether amiRSUL in Figure 2E is processed correctly and functional. RNA blots will detect miss processed species, which are inactive and common in processing mutants. Thus it is possible that in pSUC2::amiRSUL +/- you have 60% of fully active miRNA saturating AGO1 and silencing, whereas in hst mutants you have 70% but dominated by inactive misprocessed species.

-I wouldn't call a ~30% reduction measured by RNA blot "modest". It is not unusual to see these reduction levels even in mutant in canonical processing factors such as dcl1-9, tgh, drb1 (10.1371/journal.pone.0006442, 10.1073/pnas.1204915109, 10.1261/RNA.1297109, 10.1104/pp.112.193508, as a few examples). Using miR165/166 as a control for figure 2E is biased, as this is one of the few miRNAs (along with miR172 and 163) not showing changes among all

tested in the original HST report (10.1073/pnas.0405570102).

Plus you see a ~30% in this figure but later on a ~40% in figure 4 meaning that this could be variable and strong in some cases.

-Figure 3B, perhaps you have to provide a quantification here of the ratios between input/IP.

-Line 254: how this fit previous reports showing that hst mutants display impaired miRNA activity over several well-known miRNA targets? Figure 3C may also suggest an impaired activity, although it is hard to score using this promoter. You may need to test the hypothesis in the other way around. Express the amiRSUL under a set of general promoters, even inducible ones, and then test bleaching in WT and hst mutants. If HST has nothing to do miRNA activity/production, then the silencing should remain the same in both backgrounds.

-Figure 4A: please check quantification of the band in the lane "Vasculature-hst-1 (amiRSUL)". At naked eye this band is notoriously less intense than the control and its corresponding U6 loading control is the highest, 0.9 appears erroneous. I recommend expressing the quantification as a ratio of the normalized values for each background between Vasculature vs Epidermis. hst mutants display a notorious reduction of amiRSUL in vasculature than its control so it will be expected to have less movement to epidermis. For example, a quick quantification using the provided image tells me that amiRSUL "moves" 30% less in hst1 and 40% less in -25 and -26 which is similar to the "modest" 30% miRNA reduction of the mutants. Plus movement may not be linear and only "excess miRNAs" move, thus the 30% reduction in miRNA may result in a considerable % reduction in movement. I believe Figure 4B and C are more substantial evidence of impaired movement.

-L378: "This inference was further substantiated..." For years this function was challenged and disproved by quite a significant amount of evidence. After its initial discovery, HST was further confirmed as a miRNA related factor, but its function as miRNA transported was refuted in several articles (see concern 1). There is a new paper in Mol. Plant also providing evidence that hst is not shuttling, at least not massively, miRNAs from the nucleus to the cytoplasm. To be fair, it was probably published after this paper was submitted.

-Paragraph starting in line 373 is not conclusive. First, hst-3 has, like many other hst alleles, low levels of miRNAs (10.1073/pnas.0405570102) and likely normal Nucleus/cytoplasm miRNA partition as observed for different alleles (10.1073/pnas.0405570102, 10.1111/tpj.14445, 10.1038/s41477-020-0726-z), even RAN mutants shows normal N/C partition (10.1016/j.molp.2020.12.019). So I doubt this allele has any change in miRNA shuttling and even alter subcellular distribution. Authors may need to test their hypothesis in ran mutants, confirm that, opposite to RAN mutants and other hst alleles, hst-3 has altered N/C miRNA distribution, and confirm by confocal microscopy that hst-13 protein is not shuttling as proposed. Second, the portion of miRNA bound to HST (Fig. S13) is nearly undetectable to explain the amount of movement detected in other experiments. In summary, the evidence presented in this paragraph is not sufficient to attribute HST any cargo-shuttling function.

-L435: This is not probing that "nucleocytoplasmic transport enabled by the XPO5 activity of HST - which is likely required for amiRSUL movement - is dispensable, by contrast, for the mobility of SUL437 derived siRNAs produced under near-identical experimental settings". This only shows that without HST the movement is impaired, whether this is a characteristic of HST or just an indirect effect of reduced miRNAs is unclear. Actually the fact that siRNAs, which are produced by an unrelated pathway, keep moving point to a problem with miRNA biogenesis rather than movement as the cause of the detected effect. It would be a good idea to test atasiRSUL described by

Felippes 2011 as this construct require miRNA production for their biogenesis but will move as siRNAs. Thus if tasiRNA-mediated silencing movement is impaired, or reduced, in hst mutants it will tell us that impaired production of miRNAs is the cause of the reduced movement.

-I believe there is a crucial control missing all around: testing in most experiments side to side pSuc2::amiRSUL x hst with pSuc2::amiRSUL x dcl1-9/se/tgh/dr1. Most if not all experiments will likely give very similar results in both.

-Experiments shown in Figure 6D and F, are probably the more robust evidence of HST affecting movement. Given the importance of these experiments, a miRNA quantification is in order here (qRT-PCR). In any case, even when this is the strongest evidence in the paper of movement it does not imply that: a. HST participate in miRNA N/C shuttling; b. that HST is not a miRNA biogenesis factor. Their observations are compatible with HST not shuttling miRNA and participating in miRNA production, and more harmonious with current evidence of HST function.

-L536: Park et al. 2005 showed a 20% reduction in miR160 that judging by the loading control is larger. Zhu et al. 2019 showed a decrease of ~60% although in rice-hst mutants.

-L621-624 I'm afraid I have to disagree with this statement as explained before. The provided experiments do not allow such a conclusion by any mean.

-L625-627 here is the main problem with the model and discussion. They based the story in two, not substantiated, premises: a. that HST moves miRNAs out of the nucleus (a function that was repeatedly refuted by evidence), and b. that HST has nothing to do with miRNA production, a role that is likely to happen at a cell-autonomous level. The authors need to re-think their model and discussion out of those two premises.

-L644 "a step, which, we propose, is facilitated by HST" this was refuted by evidence. The arguments: a hardly detectable miRNA IP and that free miRNAs are available in the cytoplasm, are not enough to support this idea and definitively not enough to refute numerous reports showing no changes in the N/C partition in hst or ran mutants. For example, a free movement of miRNAs out of the nucleus and a weak cytoplasmic interaction with HST that triggers movement can give the same results.

-L689: Park et al. report showed miRNA fluctuation, especially in flowers compared to other tissues (a relative increment in this tissue). In this context, the new report showing an effect of HST on DCL1 recruitment to some MIRNA loci (10.1016/j.molp.2020.12.019) may also explain this phenomenon, as flowers are a potential niche for DCL3 activity, especially under deficient DCL1 activity.

-Damage 2019 found a massive pool of unbound miRNAs, if hst would be involved in shuttling it, this effect would be visible in N/C fractioning. In line 694, they mention that the lack of differential N/C miRNA accumulation in hst mutants may be explained as AGO1 loading and export can increment and compensate. In Figure 3B, the authors do not observe any change in AGO1-miRNA loading (neither artificial nor endogenous). Actually, amiRSul loading is slightly reduced in all three hst alleles. Furthermore, if this theory is correct, free miRNAs in cytoplasm should be depleted in hst mutants and nuclear AGO1 loaded miRNAs enriched. Two experiments that could be performed.

Referee #1:

This is an excellent paper that finally explain the role of HST in the miRNA pathway and reconcile all existing data. The authors show that HST (the plant XPO5 ortholog) is required to export miRNAs from emitting tissues, but not for miRNA reception or miRNA activity in recipient tissues. Their results are very clear and convincing, and what they found using a system based on an artificial miRNA applies also to endogenous miRNAs. The paper is well written and discussed, and represents a real advance in our understanding of the plant miRNA pathway.

We thank the reviewer for his kind comments and appreciation of our work. We note, however, that exactly when our manuscript was under review, a major, in-depth study conducted on *Arabidopsis* HST was published (Cambiagno et al., 2021), which essentially shattered the notion that the XPO5 function of HST is involved in miRNA nuclear export, a supposition upon which our original model was temptatively based in the first version. In the meantime, experiments requested by reviewer 2 indicate that the localization of *hst-3::GFP* remains nucleocytosolic unlike what was predicted in the yeast two hybrid system (and what we had predicted !), yet *hst-3*, as a mutation, abrogates amiRSUL movement whereas ubiquitous expression of the *hst-3* mutant protein fails to rescue this phenotype. These experiments are now included and discussed in the revised manuscript on line 454-463 and new Figure 6C. We are thus led to conclude that nucleocytosolic shuttling of HST is neither required for miRNA nuclear export (as recently reported), nor for the control of movement.

We have now revised our discussion to revoke our initial model in which HST is the nucleocytosolic transporter of miRNAs as a prerequisite to their movement. However, we found great comfort in the recent results of Cambiagno et al., (2021), which position HST at the nexus of *MIRNA* transcription and processing in a manner that does not implicate HST in miRNA processing *per se*, but would allow the protein, through its reported interaction with DCL1, to seize non-loaded miRNAs before their loading into AGO1 in the nucleus. Nuclear loading would seal their cell-autonomous fate in executing intra-cellular silencing in the silencing emitting cells only, because AGO1 is cell-autonomous. The role of HST would be to transiently channel the AGO1-unbound molecules into a non-autonomous pathway that would ultimately bring them to the cytosol and thereby enable part of this material to move to neighboring tissues. This new model is very similar, in essence, to the previous one. However, HST is no longer presented as the exporter *per se* but, rather, as a transient facilitator of the nuclear export (in a HST-independent manner) of AGO1-unbound miRNAs, which, we show, are the likely mobile entity. As to the experiments and their implications, they remain essentially unchanged so that the modified model does not denature in any perceivable manner the original findings reported in the first version. The revised model also does not alter the interpretations originally made, in the discussion, on the asymmetrical requirement for HST in the silencing-emitting but not -receiving tissues, its cell autonomy, or the lack of effect of *hst* on siRNA-, in contrast to miRNA- movement. The new entries and modifications to the original text are indicated in red to help the reviewer to navigate through the amendments required for setting the new model in relation to the recent work of Cambiagno et al., (2021). We are hopeful that the referee will agree to them and maintain his/her original enthusiasm for the amended version.

I only have a few suggestions:

The authors show the reduction of SUL mRNA accumulation in leaves of pSUC2::amiRSUL plants and in roots of wt plants grafted to pSUC2::amiRSUL scions, but not in roots of pSUC2::amiRSUL plants, which should be the control for Fig1D. Is the reduction of SUL mRNA accumulation as efficient with amiRSUL transmitted to the roots as it is with amiRSUL produced in the roots ?

We had in fact conducted the experiments and have added the results to the amended Figure 1D. They indeed confirm a cumulative effect of the locally-produced and mobile amiRSUL. We have added their description in line 154-156.

The effect of *se-1* is not convincing, and should be removed. The effect of *ago1-27* is limited, thus the authors cannot say that silencing is suppressed. Maybe they could remind the readers what is the effect of the *ago1-27* mutation.

The *se-1* panel in Fig.S1C was removed as suggested and the peculiarity of *ago1-27* is now explained with the adequate references in line 134-135. Finally, we have replaced the word "suppressed" by "reduced" on line 134.

The authors correctly report that *hst* does not suppress transgene S-PTGS (Park et al, 2005), indicating that HST is not required for the movement of siRNA, but they missed that *hst* in fact enhances transgene S-PTGS (Martinez de Alba et al, 2011). Given the competition between miRNA and siRNA for AGO1, maybe the authors could take this into account in their discussion.

The enhanced PTGS in *hst* had indeed escaped our attention and could be rationalized also in the new model if at least part of the non-loaded miRNA fraction normally channeled by HST into non-autonomy was destroyed in the nucleus, in the *hst* background; presumably this would free some AGO1 to load more siRNAs. However we did not manage to insightfully introduce this notion in the new discussion, which is already heavy both in content and concepts, so we hope the reviewer will agree with this choice.

Other minor comments:

One sentence in the introduction appears truncated: "Whether exosomes or other vesicles are used in sRNA trafficking between plant cells remains unknown, however, given that plant cells are separated by cell walls."

This sentence was not truncated but, in re-reading it, we found it particularly unclear, indeed. This section was re-written as follows (lines 107-109):

"Whether exosomes or other vesicles are used in sRNA trafficking between plant cells remains unknown, however, especially because plant cells are separated by cell walls unlikely to accommodate the passage of vesicles".

The authors should not use nomenclature like pSUC2::amiRSUL^{hen1} in the text and pSUC2::amiRSUL x hen1 in their figures (it looks like they are analyzing F1 plants). Instead they should use regular nomenclature pSUC2::amiRSUL^{hen1} in both text and figures.

We thank the referee for drawing our attention to this mistake in nomenclature, which indeed would have suggested the use of F1. This has now been changed at all relevant places in the text and figures.

Referee #2:

1. This manuscript describes HASTY's identification and characterization as a critical player during non-cell-autonomous miRNA movement, both at short and long-distance, in plants. Using a genetic approach, the authors found several alleles of *hst* in a screening designed to identify mutants deficient in miRNA movement. Later on, the authors showed that the molecules moving are likely mature miRNA rather than the precursors. They also found that it is unlikely that AGO1 bound miRNAs are the moving entity pointing to the free cytoplasmic portion of miRNAs as the best candidate for non-cell-autonomous effect. Similarly, they found that HST itself is not moving out of the cell and thus, whatever effect it has on miRNA movement is cell-autonomous. Methodologically the paper is solid, and the experiments are well designed and performed.

We thank the reviewer for acknowledging the quality of the design and execution of the experiments and have taken on board his/her criticisms regarding some of the interpretations, as detailed throughout this rebuttal. Note that the changes made to the original text are now indicated in red to facilitate the reviewer's perusal of the amended text.

2. Even when I only have a few concerns about the experimental part itself, I believe the papers' proposed model and conclusions are biased, not supported by the evidence provided and unnecessarily opposed to the current knowledge of HST function. For example, one of the central premises in their construction of the story is that miRNA production is not affected in *hst* mutants. Probably the authors took this path, as a reduction in miRNA production will largely explain most of the results and observations. The provided evidence does not refute the impact of *hst* mutation on miRNA production. The authors need to better fit their model and results to existing evidence that suggests that HST significantly impacts miRNA accumulation. Some evidence (especially figure 6) is strong enough to demonstrate an effect of HST on miRNA movement, even if it also has a role in miRNA accumulation.

We are very sorry that the wording of our initial manuscript could have been interpreted as advocating a biogenesis-exclusive, as opposed to -inclusive (as intended), function for HST in miRNA movement. This point is particularly important given the recent results of Cambiagno et al., (2021), which were published as the manuscript was under review. We are sorry for the misunderstanding as it was not our intention to separate biogenesis from movement. In fact, both are likely intimately linked due to the very special positioning and action of HST in the nuclear miRNA biogenesis pathway as recently revealed by Cambiagno et al., (2021). To take into account this point more clearly, we have reworded the manuscript all throughout in a manner reflecting the two intertwined effects of HST. In the case of *amiRSUL*, for instance, we have reworded our accounts of the leaf-peeling and grafting experiments by specifically stating that while *hst* unquestionably decreases the levels of *amiRSUL* in silencing-emitting tissues, this decrease is consistently and substantially higher in silencing-receiving the in -emitting tissues, arguing that, ***in addition to its role in amiRSUL biogenesis, HST also controls its movement.*** This wording is now explicit all throughout the revised version as well as in the abstract.

The revised discussion (prompted by the criticisms evoked in point 3 below) also now makes a strong case for the role of HST in biogenesis being at the very core of its function in miRNA movement. That said, there are several cases, including with endo-miRNAs, where we did not observe any overt changes in the levels of miRNAs in *hst* silencing-emitting tissues, but this does not conflict with the general role for HST in biogenesis being more visible with some, unlike other, endo-miRNAs for reasons that are possibly explained, precisely, by its role in movement, as evoked in the discussion. Our reanalysis of the sRNA seq data publicly available from Cambiagno et al., (2021) confirms, as they reported, that approx. 30% of *Arabidopsis* miRNA display altered (most often downregulated) levels in *hst-15*. However, and as is also visible in the supp table S3 of the original publication, miR160, miR395 and miR165 (the mobile endo-miRNAs studied in our manuscript) are not part of this cohort (see point 6.). We refer to the new Fig.S10 to emphasize this point when necessary in the text.

3. Similarly, and in the absence of a molecular mechanism that explains how HST control miRNA movement, the authors rely on the assumption that HST has a significant role in miRNA shuttling from nucleus to cytoplasm that impacts miRNA movement. Evidence provided by numerous authors, including themselves in previous papers and this one, has refuted a potential role of HST on miRNA shuttling (see below). Authors need to either provide convincing evidence of the premises mentioned above or present a less speculative model based on available data. A recent paper offers new evidence on HST functions and needs to be considered here; probably the authors were not aware of this publication by the submission time as it is quite new.

As explained in our response to point 2, above, we have indeed read the recent paper by Cambiagno et al., (2021) and agree with the conclusion that HST plays no or little role in miRNA nuclear export. In fact, this is further supported by our analysis of *hst-3:GFP* subcellular localization, as requested by referee 2 (point 9), which shows that there is no correlation between HST's nucleo-cytosolic shuttling and its role in movement. This has prompted us to abandon this notion from our initial model in the discussion, in agreement with the referee's opinion. We now focus the model on linking the very interesting position of HST at the nexus of *MIRNA* transcription and miRNA

processing (as defined in Cambiagno et al., (2021)) with the probable capacity for the protein to transiently seize (or “snatch”) a portion of miRNAs before their loading into AGO1. Loading would indeed seal their fate as cell-autonomous entities as explained in our study, which argues strongly against the movement of AGO1:miRNA complexes given the cell autonomy of AGOs and AGO1 in particular. The portion of non-loaded miRNAs seized by HST (likely facilitated by its miRNA-processing independent interaction with DCL1) would then be amenable to movement after their delivery into the cytosol by an as yet unidentified HST-independent mechanism. Thus, in this revised model, HST would merely transmit the unloaded miRNAs to a non-cell autonomous pathway. The discussion has been considerably altered and shortened to take into account these considerations in a manner now emphasizing the intimacy between biogenesis and movement. Note that the revised model does not alter in any manner the interpretations originally made in the discussion on the asymmetrical requirement for HST in the silencing emitting but not receiving tissues only, its cell autonomy, or the lack of effect of *hst* on siRNA-, in contrast to miRNA- movement.

-L42-45: it would be a good idea to introduce in this sentence also CRD1 (10.1111/tpj.14445), HASTY ortholog in rice as its molecular features and effects upon mutation are shared with Arabidopsis.

The referee is correct and this was an oversight on our part. We have now added the reference in the revised introduction on lines 46-47 and 54-56. We note also that *crd1*, despite its effect on miRNA levels, has little effect on pri/pre-miRNA processing as was shown recently by Cambiagno et al., (2021) and verified by us (amended Supp data S16B).

4. L47, a non-affected miRNA Nuclear/Cytoplasmic partition in different *hst* alleles was also observed at least by (10.1038/s41477-020-0726-z) and (10.1016/J.Molp.2020.12.019) in Arabidopsis and by (10.1111/tpj.14445) in rice. It was also observed in a previous report of the authors (10.1016/j.molcel.2018.01.007, Figure 4C) where they found no over-accumulation of nuclear miRNAs in *hst* alleles by FACS. Even when you can argue that in the absence of HST miRNA can be loaded in AGO1 and exported by this mechanism compensating the lack of HST, this idea is refuted by this article when you showed AGO1-miRNA loading does not increase in *hst* mutants (figure 3B).

As explained in our response to point 3, above, we have now abandoned the criticized idea and have added the references cited by the referee already in the introduction to further emphasize that HST does not control miRNA nuclear export.

-L169-178: Can you dismiss this possibility by measuring pri-miRSUL content in the phloem, as done in figure 1F for mature amiRSUL, using aphids?

This is a good suggestion and we had indeed tried the suggested RT-qPCR experiment before submission of the first version. However, the results have never been convincing because the signal we obtained for aphids fed on WT plants was the same as that obtained for aphids fed on *pSUC2:amiRSUL* plants, identifying it as background, essentially.

-Please change EXPORTIN5 by HASTY, which is the correct name in Arabidopsis, in the title. If you want to mention EXP5 to highlight its orthology with the metazoan gene you can go for: HASTY, the Arabidopsis ortholog of EXPORTIN5, regulates the movement of microRNAs

This has been done.

-L125-127: perhaps it is better to use the word impaired or reduced instead of suppressed, as bleaching is still observable in several of these mutants.

We have now made this change, which was also requested by referee 1 (Line 133).

-L198: I'm curious why *hst-25*, which is very similar regarding the mutation to *hst-1*, does not share a similar phenotype even when the protein is undetectable in both alleles (Figure 2B). Have you cloned and sequenced the two lower bands observed in figure S5 for *hst-25* to address if some of these are the correctly spliced mRNA or a version retaining the ORF?

We have followed the suggestion of the referee and cloned the bands detected in figure S5, for *hst-25*, but also *hst-1*. The outs are now disclosed in the amended figure and indicate that the two lower bands in *hst-25* correspond to

RT-PCR products produced from (i) the normally spliced mRNA coding for a WT HST protein, and (ii) an alternatively spliced shorter mRNA, which actually remains potentially coding for a putative protein truncated of 16 amino acids. The lower band in *hst-1* corresponds to an RT-PCR product derived from the same alternatively spliced mRNA, giving rise to a similar truncation of 16 amino acids in encoded HST protein. Although the WT and/or truncated HST proteins in *hst-1* and *hst-25* remain below detection levels in western analysis, *hst-25* still expresses a normally-sliced, remnant mRNA coding for a WT protein, likely explaining the reduced effect of the mutation (as for *hst-26*) on the developmental phenotype as compared to *hst-1*.

This is now explained in the main text (lines 204-212) when we refer to the molecular features of each of the *hst-1*, *hst-25* and *hst-26* mutations.

The authors argue that *hst* mutants have no impaired miRNA production in part based on the fact that *hst-25* and -26 still have some degree of vein chlorosis. Although this is true, the reduction is notorious and compatible with the previously reported decrease in miRNA production. As a matter of fact, the lack of chlorosis observed in these plants (figure 2C) is even stronger than in canonical biogenesis mutants such as *hyl1* and *se*, or miRNA turnover mutants such as *hen1* (Figure S1C). The complete loss of chlorosis in *hst-1* (fig 2C) also point in the direction of a robust cell-autonomous effect in the mutant on miRNA production.

We have now completely re-worded this section (line 243-262) in accordance to point 5 raised by the reviewer. See our response below.

5. On the paragraph starting in line 214: It is impossible to address whether the amiRSUL in figure 2F is enough to saturate the system and remain active. Conversely, is not clear here whether amiRSUL in Figure 2E is processed correctly and functional. RNA blots will detect miss processed species, which are inactive and common in processing mutants. Thus it is possible that in pSUC2::amiRSUL +/- you have 60% of fully active miRNA saturating AGO1 and silencing, whereas in *hst* mutants you have 70% but dominated by inactive misprocessed species.

We agree with the referee that, without more data, it is difficult to strictly compare amiRSUL activity in *hst* to that in WT. We have now integrated the various possibilities raised by the reviewer and tested each of them meticulously in a new section between line 243-262 and 274-290. In particular, an analysis of the *hst-15* sRNA seq data from Cambiagno et al., (2021) did not show evidence of increased endo-miRNA tailing/trimming as assessed recently in Giudicatti et al., (2021) or misprocessing as analyzed by us in Iki et al, (2018). Based on the fact that neither the AGO1-loading efficacy nor the silencing activity of amiRSUL are overtly affected, we tentatively attribute (but no longer affirm this) the drastic difference in vein-centered silencing phenotypes between pSUC2::amiRSUL hemizygotes with the WT background versus pSUC2::amiRSUL homozygotes with the *hst* background, to a potential role for HST in controlling amiRSUL movement *in addition* to controlling its biogenesis. We also carefully explain, at the end of this new section, that measuring amiRSUL movement indirectly by assessing its activity in recipient tissues (via chlorosis) is *too ambiguous to conclude definitively*, which calls for a direct assessment of its physical movement via epidermal peeling and grafting.

6. I wouldn't call a ~30% reduction measured by RNA blot "modest". It is not unusual to see these reduction levels even in mutant in canonical processing factors such as *dcl1-9*, *tgh*, *drb1* (10.1371/journal.pone.0006442, 10.1073/pnas.1204915109, 10.1261/RNA.1297109, 10.1104/pp.112.193508, as a few examples). Using miR165/166 as a control for figure 2E is biased, as this is one of the few miRNAs (along with miR172 and 163) not showing changes among all tested in the original HST report (10.1073/pnas.0405570102). Plus you see a ~30% in this figure but later on a ~40% in figure 4 meaning that this could be variable and strong in some cases.

We have now removed statements qualifying the reductions in amiRSUL seen in the various experiments. Also for the sake of objectivity, we have now shown in the northern in the original Figure 2E (now Figure 3A) various hybridizations made for many additional endo-miRNAs. Some are significantly reduced, others much less, if at all, or even occasionally up-regulated, in agreement with the *hst-15* sRNA seq analysis in Cambiagno et al., (2021). We now acknowledge better, we believe, that *hst* does have an undeniable effect on amiRSUL levels in silencing emitting tissues. However both the peeling and grafting experiments, which are now shown with multiple replicates (see amended Figures 4; S12 and our response to point 8.2), consistently indicate that the reduction seen in silencing-recipient tissues is substantially more pronounced than that seen in -emitting tissues. We thus conclude that HST likely controls amiRSUL movement *in addition* to controlling its biogenesis, a phrasing that does not oppose the two processes as is also further emphasized now in the revised discussion advocating, on the contrary, a link between the two.

-Figure 3B, perhaps you have to provide a quantification here of the ratios between input/IP.

This has been done, although it is difficult to find a suitable normalization of RNA signals in IP fractions. According to the western analyses of AGO1 in the input, unbound and IP fractions, we could estimate that the immunoprecipitating beads were saturated in AGO1, leading to a direct normalization of the RNA extracted from them.

7. Line 254: how this fit previous reports showing that *hst* mutants display impaired miRNA activity over several well-known miRNA targets? Figure 3C may also suggest an impaired activity, although it is hard to score using this promoter. You may need to test the hypothesis in the other way around. Express the amiRSUL under a set of general promoters, even inducible ones, and then test bleaching in WT and *hst* mutants. If HST has nothing to do miRNA activity/production, then the silencing should remain the same in both backgrounds.

Line 254 referred to unusual experiments disclosed in the original figure 3E (now Figure 3F), which report the activity of amiRSUL processed in the WT background (the scion) in an *hst* mutant recipient tissue (the rootstock). It is distinct from looking at the activity of a miRNA processed in the *hst* background. Regarding the targeting defects of *hst*, we are now referring, in line 276-280, to the RNA seq analysis results from Cambiagno et al., (2021), which we have now crossed with the total set of annotated miRNA target transcripts in *Arabidopsis* (revised Fig.S10C). The analysis indicates that 20 out of 249 (8%) miRNA target transcripts show significantly increased levels in *hst-15* versus WT plants (Fig.S10C), of which 8 (3%) are targets of miRNAs reported as being significantly down-regulated in the same study and as confirmed by us (Cambiagno et al., 2021; Fig.S10A, S10D). Altogether, the lack of effect of *hst* on the activity of amiRSUL synthesized in a WT background is, in our view, in line with the results of the above RNA seq analysis even though it was carried out in a distinct *hst* allele.

8.1. Figure 4A: please check quantification of the band in the lane "Vasculature-*hst-1* (amiRSUL)". At naked eye this band is notoriously less intense than the control and its corresponding U6 loading control is the highest, 0.9 appears erroneous. I recommend expressing the quantification as a ratio of the normalized values for each background between Vasculature vs Epidermis.

The referee is correct: the 0.9 value is not representative of the experiment shown, as it was meant to reflect a mean value of four independent replicates of these peeling experiments, two of which are now shown in the amended Figure 4A and the two remaining in Figure S12A, given their importance here. We apologize that we had not specified in the original figure legend that the values presented corresponded to a mean value. We have now indicated the signal quantification relative to HST for each *hst* allele on each individual replicate and have also expressed them as a final vasculature/epidermis ratio as requested and now depicted in Figure 4B.

8.2. *hst* mutants display a notorious reduction of amiRSUL in vasculature than its control so it will be expected to have less movement to epidermis. For example, a quick quantification using the provided image tells me that amiRSUL "moves" 30% less in *hst1* and 40% less in -25 and -26 which is similar to the "modest" 30% miRNA reduction of the mutants. Plus movement may not be lineal and only "excess miRNAs" move, thus the 30% reduction in miRNA may result in a considerable % reduction in movement. I believe Figure 4B and C are more substantial evidence of impaired movement.

All signals on the original phosphorimager-generated data were carefully quantified using the BioRad ImageLab software. As is visible in each of the four replicates of the experiment (Figure 4A, S12A), the reduction in amiRSUL levels in the silencing-recipient epidermis consistently exceeds that seen in the silencing-emitting vasculature. So, we respectfully disagree with the comment made here by the referee. In fact, after deducing the effect of *hst* on amiRSUL biogenesis in the vasculature, its epidermal movement *stricto sensu* is reduced by 59%, 68% and 64% in *hst-1*, *hst-25* and *hst-26* compared to WT background. The average (64%) is fully in line with that observed for the reduction of endo-miR160 movement (63%) in the peeling experiments involving *hst-1* (Figure 7C-D), which the reviewer had no quarrel with in his original assessment.

The original panels in Fig.4B (now moved in Fig.S12B as replicates of new panel 4C) reporting the graft transmission of amiRSUL provide consolidating evidence that the long-distance movement of amiRSUL (after deducing the effects of *hst* on biogenesis in the scions) is impaired by 61% on average by *hst-1* and *hst-25*. This is in line with the reduction in cell-to-cell movement evoked above, but also with the impaired graft-transmission of endo-miR395 (54%; Figure 7C-D), which, again, the reviewer had no quarrel with in his original assessment.

-L378: "This inference was further substantiated..." For years this function was challenged and disproved by quite a significant amount of evidence. After its initial discovery, HST was further confirmed as a miRNA related factor,

but its function as miRNA transport was refuted in several articles (see concern 1). There is a new paper in Mol. Plant also providing evidence that *hst* is not shuttling, at least not massively, miRNAs from the nucleus to the cytoplasm. To be fair, it was probably published after this paper was submitted.

Taking the referee's comments and our analysis of *hst3:GFP* (see below, point 9) into account, we have now significantly reworded this section and subheading (lines 430-463). Our conclusions are now that there is indeed no strong evidence that the shuttling of HST is involved in amiRSUL movement, in agreement with the reviewer's opinion. See also our answers to points 3, 4, 9 here.

9. Paragraph starting in line 373 is not conclusive. First, *hst-3* has, like many other *hst* alleles, low levels of miRNAs (10.1073/pnas.0405570102) and likely normal Nucleus/cytoplasm miRNA partition as observed for different alleles (10.1073/pnas.0405570102, 10.1111/tpj.14445, 10.1038/s41477-020-0726-z), even RAN mutants shows normal N/C partition (10.1016/j.molp.2020.12.019). So I doubt this allele has any change in miRNA shuttling and even alter subcellular distribution. Authors may need to test their hypothesis in ran mutants, confirm that, opposite to RAN mutants and other *hst* alleles, *hst-3* has altered N/C miRNA distribution, and confirm by confocal microscopy that *hst-3* protein is not shuttling as proposed. Second, the portion of miRNA bound to HST (Fig. S13) is nearly undetectable to explain the amount of movement detected in other experiments. In summary, the evidence presented in this paragraph is not sufficient to attribute HST any cargo-shuttling function.

As stated, we have conducted the requested subcellular localization experiments (Fig.6C) and they do not provide support for the notion previously put forward by us in our original model (although they stated at the time that it was speculative and based merely on an *inferred* function of HST in miRNA export). We have accordingly reformulated our model to take into account the biogenesis role for HST as correctly evoked multiple times by the referee.

10. L435: This is not probing that "nucleocytoplasmic transport enabled by the XPO5 activity of HST - which is likely required for amiRSUL movement - is dispensable, by contrast, for the mobility of SUL437 derived siRNAs produced under near-identical experimental settings". This only shows that without HST the movement is impaired, whether this is a characteristic of HST or just an indirect effect of reduced miRNAs is unclear.

We have now removed the statement on XPO5 activity and simply provide an account of the lack of effect of *hst* on the SS siRNAs (Line 484-486).

Actually the fact that siRNAs, which are produced by an unrelated pathway, keep moving points to a problem with miRNA biogenesis rather than movement as the cause of the detected effect.

This is true and does not contradict our initial views. Both the distinct location and machineries involved in siRNA, as opposed to miRNA, biogenesis are now acknowledged in the revised discussion as possible causes for the lack of effect of *hst* on siRNA mobility (Line 758-759).

It would be a good idea to test atasiRSUL described by Felipe 2011 as this construct requires miRNA production for their biogenesis but will move as siRNAs. Thus if tasiRNA-mediated silencing movement is impaired, or reduced, in *hst* mutants it will tell us that impaired production of miRNAs is the cause of the reduced movement.

This is a good idea in principle. The atasiRSUL system evoked by the referee involves the AGO1-mediated cleavage of an artificial SUL-precursor based on the *TAS1a* non-coding transcript targeted by endogenous miR173. However, as shown in the RNA blot provided here for the referee's perusal, none of the *hst* alleles tested by us alters miR173 accumulation and nor do they alter, likewise, TAS1-siRNAs production, indicating unaltered AGO1:miR173 activity. We conclude, as is the case for other miRNAs and their targets (Supp S10; Cambiagno et al., (2021)), that *hst* does not compromise the miR173-TAS1 network, making it unlikely that the suggested atasiRSUL experiment would be conclusive.

11. I believe there is a crucial control missing all around: testing in most experiments side to side pSuc2::amiRSUL x *hst* with pSuc2::amiRSUL x *dcl1-9/se/tgh/dr1*. Most if not all experiments will likely give very similar results in both.

Given that all genotypes must be grown in parallel due to the biological variations observed in the effects of *hst* on amiRSUL and, to a lower extent endo-miRNA levels, the reviewer's request would imply the reproduction of nearly all experiments and their replicates, which is impossible given the time allocated for our revisions. Nonetheless, we took onboard the argument that mutations in any miRNA biogenesis factor, and not *HST* in particular, would lead to defects diagnosed as impaired movement whereas they would, in fact, merely reflect impaired biogenesis in the silencing emitting tissue.

As is now visible in panel C of the amended Figure 4, we had in fact conducted grafting experiments allowing us to test the above possibility with two well-established, strong mutations affecting miRNA biogenesis at two levels: *hyl1-2*, reducing pri/pre-miRNA processing and *hen1-6*, impairing post-processing miRNA stability in the nucleus. While these experiments were not included in the submitted version, we had also performed, for other purposes, a similar analysis with *hen1-14* isolated in our screen, as presented in Figure 1G, and with *hyl1-2* as presented in Figure S3A. As is now discussed in the main text (Lines 311-328), we have confirmed that amiRSUL levels in *pSUC2::amiRSUL^{hyl1-2}* or *pSUC2::amiRSUL^{hen1-6/hen1-14}* scions were substantially more reduced compared to those in *pSUC2::amiRSUL^{hst-1}* or *pSUC2::amiRSUL^{hst-25}* scions. However, in stark contrast to the 69% and 53% movement reductions observed with the *hst-1* and *hst-25* scions, movement of the residual amiRSUL derived from *pSUC2::amiRSUL^{hyl1-2}* or *pSUC2::amiRSUL^{hen1-6/hen1-14}* scions was not significantly reduced (*hen1* backgrounds) or reduced by only 20% (*hyl1-2* background), as compared to the WT- background (Fig.4D). These results indicate that decreasing amiRSUL biogenesis in silencing-emitting tissues – even up to 71% as in *hen1* or *hyl1* does not suffice, *per se*, to cause a strong deficit in movement, efficacy further emphasizing the singular effect of *hst* on this process.

12. Experiments shown in Figure 6D and F, are probably the more robust evidence of HST affecting movement. Given the importance of these experiments, a miRNA quantification is in order here (qRT-PCR).

As requested, we have now conducted stem-loop qRT-PCR quantification in all experiments involving endo-miRNA mobility, in the peeling and grafting settings. These quantifications are the foundations for the bar charts (epidermis/vasculature and rootstock/scion ratios) now depicted in Figure 7B and 7D. The results are in line with those obtained by band intensity quantification in the northern analyses.

In any case, even when this is the strongest evidence in the paper of movement it does not imply that:

a. HST participate in miRNA N/C shuttling;

We have now abandoned this line of argumentation in the introduction, results and discussion sections of the revised manuscript. See our answers to point **3** and **9** as well.

b. that HST is not a miRNA biogenesis factor. Their observations are compatible with HST not shuttling miRNA and participating in miRNA production, and more harmonious with current evidence of HST function.

As discussed throughout the rebuttal, we have emphasized as much as possible that the effect of HST on movement is not achieved in contradiction, but *in addition*, to its effects on biogenesis. Furthermore, the amended discussion now makes a strong case for a mechanistic link between the position of SHT in nuclear miRNA biogenesis pathway.

-L536: Park et al. 2005 showed a 20% reduction in miR160 that judging by the loading control is larger. Zhu et al. 2019 showed a decrease of ~60% although in rice-*hst* mutants.

We have now removed these statements on individual miRNAs levels measured sporadically on case-by-case analyses in various tissues and conditions. After all, they are of rather limited value. We now only refer to the global changes analyzed genome-wide in the sRNA seq conducted in triplicates in *hst-15* versus WT *Arabidopsis* ((Cambiagno et al., (2021); FigS.10)). We also refer to the many replicates of peeling and grafting experiments which, in addition to other, new results, now make a compelling case, we believe, for HST-dependent movement of amiRSUL and endo-miRNAs.

-L621-624 I'm afraid I have to disagree with this statement as explained before. The provided experiments do not allow such a conclusion by any mean.

The statement has been removed and the original line of argumentation abandoned as explained throughout this rebuttal.

13. L625-627 here is the main problem with the model and discussion. They based the story in two, not substantiated, premises:

a. that HST moves miRNAs out of the nucleus (a function that was repeatedly refuted by evidence),

We have amended the text all throughout to remove such statements. See our answers to point **3, 4, 8.2, 9.**

and

b. that HST has nothing to do with miRNA production, a role that is likely to happen at a cell-autonomous level. The authors need to re-think their model and discussion out of those two premises.

We are very sorry that our original efforts to implicate HST in miRNA movement were misconstrued as a means to negate or diminish its effect on miRNA biogenesis. This was certainly not our intention. We believe we have now rewritten our text so that it is understood that movement is an additional function of HST, which, as explained moreover in the discussion, is likely deeply intertwined with the position and action of HST in the nuclear miRNA biogenesis pathway. See our answer to points as well **8.2, 9, 10** among others.

-L644 "a step, which, we propose, is facilitated by HST" this was refuted by evidence. The arguments: a hardly detectable miRNA IP and that free miRNAs are available in the cytoplasm, are not enough to support this idea and definitively not enough to refute numerous reports showing no changes in the N/C partition in hst or ran mutants. For example, a free movement of miRNAs out of the nucleus and a weak cytoplasmic interaction with HST that triggers movement can give the same results.

As indicated earlier in the rebuttal, we have performed the subcellular localization analysis of hst-3 :GFP and the data provide not support for nucleo-cytosolic shuttling of HST being involved in the control of miRNA movement. Neither is HST involved in miRNA nuclear export, as pointed out by the referee. We believe the weak signal detected in the IP might reflect a very transient miRNA:HST interaction that would be consistent with the mere transmission of non-loaded miRNAs to a non-autonomous pathway, as opposed to HST being actively involved in export. This interpretation potentially agrees with the use of crosslinking by Cambiagno et al., (2021) to detect robust signal in their HST RIP experiments. We have now included these observations in our revised discussion (line 724-728).

-L689: Park et al. report showed miRNA fluctuation, especially in flowers compared to other tissues (a relative increment in this tissue). In this context, the new report showing an effect of HST on DCL1 recruitment to some MIRNA loci (10.1016/j.molp.2020.12.019) may also explain this phenomenon, as flowers are a potential niche for DCL3 activity, especially under deficient DCL1 activity.

This is indeed an interesting idea. However, given that we no longer refer specifically to quantitative variations of miRNAs observed in previous case-by-case studies, we do not think that the above explanation is needed anymore. See also our last answer, in this rebuttal, to the various issues raised in point **12.**

-Damage 2019 found a massive pool of unbound miRNAs, if hst would be involved in shuttling it, this effect would be visible in N/C fractioning. In line 694, they mention that the lack of differential N/C miRNA accumulation in hst mutants may be explained as AGO1 loading and export can increment and compensate. In Figure 3B, the authors do not observe any change in AGO1-miRNA loading (neither artificial nor endogenous). Actually, amiRSul loading is slightly reduced in all three hst alleles. Furthermore, if this theory is correct, free miRNAs in cytoplasm should be depleted in hst mutants and nuclear AGO1 loaded miRNAs enriched. Two experiments that could be performed.

We agree with the reviewer that it was an incongruity not easily explained by our previous model. A situation in which HST is a mere transient "snatcher" of unload miRNAs is better accommodated because it predicts that only the nuclear pool AGO1 as opposed to the total cellular pool thereof (which is involved in the IPs of figure 3B) would specifically display more loading with at least some miRNAs, but not others. We had generated potentially satisfactory molecular tools to address this point in our lab (Bologna et al. 2018) and are in the process of bringing these tools in adequate genetic backgrounds, yet the timing would exceed the allowance for revision here. We discuss, this point in the new discussion, nonetheless, along the lines outlined by the referee here.

--- End of rebuttal ---

Thank you for submitting a revised version of your manuscript. Your revised study has now been seen by both original referees, who find that their concerns have been addressed and recommend publication of the manuscript. Therefore, I would like to invite you to address the remaining editorial issues before I can extend the official acceptance of the manuscript.

Please let me know if you have any further questions regarding any of these points. You can use the link below to upload the revised files.

Referee #1:

I am totally satisfied with the answers of the authors to my comments and to the comments of the other reviewer. It is nice that the paper by Cambiagno et al appeared during the reviewing process so that the authors got a chance to revise their model and draw conclusions that fit with these new data.

Referee #2:

I'm very happy with this new version of the manuscript. The authors reshaped the manuscript to better fit the current knowledge about HASTY and provided new solid evidence or clear arguments to answer my concerns. I appreciate the effort the authors made to answer my questions and reshape the manuscript. I believe this article is now a solid piece of literature that not only provide new evidence about the miRNA pathway but also open the door to many exciting questions.

The authors performed the requested editorial changes.

Editor accepted the revised manuscript.

Corresponding Author Name:

Journal Submitted to:

Manuscript Number: